# Sphingomyelin-derived nanovesicles for the delivery of the IDO1 inhibitor epacadostat enhance metastatic and post-surgical melanoma immunotherapy

Zhiren Wang [1], Wenpan Li [1], Yanhao Jiang [1], Tuyen Ba Tran[1], Leyla Estrella Cordova [1], Jinha Chung[1], Minhyeok Kim [1], Georg Wondrak [1,2], Jennifer Erdrich [3] & Jianqin Lu [1,2,4,5] ✉

Epacadostat (EPA), the most advanced IDO1 inhibitor, in combination with PD-1 checkpoint inhibitor, has failed in a recent Phase III clinical trial for treating metastatic melanoma. Here we report an EPA nanovesicle therapeutic platform (Epacasome) based on chemically attaching EPA to sphingomyelin via an oxime-ester bond highly responsive to hydrolase cleavage. Via clathrin-mediated endocytosis, Epacasome displays higher cellular uptake and enhances IDO1 inhibition and T cell proliferation compared to free EPA. Epacasome shows improved pharmacokinetics and tumour accumulation with efficient intratumoural drug release and deep tumour penetration. Additionally, it outperforms free EPA for anticancer efficacy, potentiating PD-1 blockade with boosted cytotoxic T lymphocytes (CTLs) and reduced regulatory T cells and myeloid-derived suppressor cells responses in a B16-F10 melanoma model in female mice. By co-encapsulating immunogenic dacarbazine, Epacasome further enhances anti-tumor effects and immune responses through the upregulation of NKG2D-mediated CTLs and natural killer cells responses particularly when combined with the PD-1 inhibitor in the late-stage metastatic B16-F10-Luc2 model in female mice. Furthermore, this combination prevents tumour recurrence and prolongs mouse survival in a clinically relevant, post-surgical melanoma model in female mice. Epacasome demonstrates potential to synergize with PD-1 blockade for improved response to melanoma immunotherapy.

Immune checkpoint blockade (ICB) therapy targeting cytotoxic T-lymphocyte-associated protein 4 (CTLA-4) and programmed death protein 1 (PD-1)/programmed death-ligand 1 (PD-L1) has garnered breakthrough therapeutic outcomes in the realm of cancer treatment[1–4]. Despite the tremendous therapeutic potential of ICB, it only benefits a select portion of patients[5–7]. Even in immunogenic melanoma, the response rate to ICB is less than 40%[8–10]. To make the ICB more effective for cancer therapies, there has been widespread interest in developing combination immunotherapeutic platforms that target different immunosuppressive pathways simultaneously or

[1]Skaggs Pharmaceutical Sciences Center, Department of Pharmacology & Toxicology, R. Ken Coit College of Pharmacy, The University of Arizona, Tucson, AZ 85721, USA. [2]NCI-designated University of Arizona Comprehensive Cancer Center, Tucson, AZ 85721, USA. [3]Department of Surgery, Division of Surgical Oncology, The University of Arizona College of Medicine, Tucson, AZ 85721, USA. [4]BIO5 Institute, The University of Arizona, Tucson, AZ 85721, USA. [5]Southwest Environmental Health Sciences Center, The University of Arizona, Tucson, AZ 85721, USA. ✉e-mail: lu6@arizona.edu

combining the use of strategies that can turn the immunologically "cold" tumours into "hot" ones or make tumours more immunogenic, aiming to achieve improved patient response and extended survival. Compared to monotherapy, combination of the PD-1 inhibitor (Nivolumab) and the CTLA-4 inhibitor (Ipilimumab) increased response rates but was associated with severe treatment-related adverse events (TrAEs; 33-55%) and immune-related adverse events (irAEs; 40–45%)[7,9,11,12]. Recently, combining inhibitors of indoleamine 2,3-dioxygenase 1 (IDO1) enzyme with ICB have been extensively investigated in various clinical trials[11,13–18].

IDO1 is another important endogenous immune regulator, which catalytically converts L-tryptophan (Trp) to L-kynurenine (Kyn). Induced by interferon-γ (IFN-γ) and other proinflammatory signals, IDO1 is mainly expressed by tumour and dendritic cells and macrophages within the tumour microenvironment (TME)[19,20]. The depletion of cellular Trp along with the generation of downstream Kyn induced by IDO1 can activate various immunosuppressive cells (regulatory T cells (Tregs)[21], tumour-associated macrophages (TAMs)[22], and myeloid-derived suppressor cells (MDSCs)[23] while rendering the anergy and apoptosis of effector T cells[21,24,25], yielding immunosuppression in tumours. Moreover, the upregulation of IDO1 has been linked with poor prognosis in cancer patients[26,27]. Thus, therapeutically targeting IDO1 may represent a promising approach for improved treatment of diverse cancers, particularly when combined with ICB.

Various IDO1 inhibitors have been developed and demonstrated viable clinical efficacy against a variety of cancers[16,28]. Among them, epacadostat (EPA) is a highly potent and selective IDO1 inhibitor, which can effectively reduce Trp metabolism, entailing increased activation and maturation of dendritic cells, and enhanced proliferation of effector T cells and natural killer cells (NKs), as well as attenuated Tregs expansion[13,29–31]. These modulations switch the immunosuppressive tumour microenvironment (TME) to an immunogenic condition that enables robust antitumour immunity[29]. The combination of EPA with ICB significantly enhanced tumour inhibition compared to monotherapy in preclinical melanoma models[32]. Early Phase clinical trials (I and II) showed that EPA had significant IDO1 inhibitory effects with good safety profile[33,34]. Nevertheless, in a recent Phase III clinical study (NCT02752074), the combination of EPA with PD-1 inhibitor (Keytruda) failed to produce progression-free survival benefit over Keytruda alone in patients with unresectable or metastatic melanoma[14,35]. While many factors have been accounted for this negative result (e.g, experimental designs, dosing regimen, etc)[14,15,36], the extremely poor pharmacokinetics (within 7 min, ~96.3% of EPA was eliminated following intravenous (IV) administration and the tumour uptake rate was ~0.1%/g; and the maximum blood dose was ~0.35% for oral administration) is rarely criticised and evaluated[33,37,38]. We posit that a nanotechnology-enabled therapeutic delivery of EPA could improve its PK and tumour targeting, therefore, resulting in enhanced combination therapy with ICB.

Previously, we established a sphingomyelin (SM)-derived camptothecin nanovesicle delivery system, which markedly increased blood circulation time and tumour delivery, rendering fortified antitumour effects[39]. Piggybacked by the prior success, we queried whether we could develop an EPA nanovesicle using the SM-conjugation strategy for improved therapeutic delivery for EPA. SM is a naturally occurring sphingolipid in mammalian cell membranes[40,41]. Herein, we attach EPA to the hydroxyl group of SM via a highly tumour-sensitive oxime-ester linkage[42,43]. SM-EPA readily forms liposomal nanovesicles (Epacasome) in aqueous medium driven by the amphiphilicity of SM, which heighten intracellular uptake, enhance IDO1 inhibitory activity and boost T-cell proliferation in vitro as compared to free EPA. Epacasome prolongs the blood circulation half-life by 31.5-fold, increases the area under curve by 153.2-fold and enhances the tumour delivery by 46.3-fold. Epacasome also exhibits efficient and timely drug release and penetrates the tumours deeply and uniformly. Taken together, these maneuvers significantly improve the biophysicochemical properties and greatly endow Epacasome with the ability to synergise anti-PD-1 therapy for further enhanced tumour reduction, prolonged mouse survival, as well as increased IDO1 suppression and antitumour CTLs responses in a B16−F10 melanoma model without overt systemic toxicities.

Dacarbazine (DTIC) is a chemotherapeutic agent widely used in treating melanoma that works by the upregulation of NKG2D ligands on tumour cells, which activates NKs and CTLs, thereby eliciting antitumour immune responses. To elucidate whether the combination of DTIC with EPA would further potentiate the anti-PD-1 ICB, we use Epacasome to co-deliver DTIC.

Here, we show that DTIC-laden Epacasome has stronger tumour inhibition over co-administration of DTIC/Lipo-SM/Chol and Epacasome, and it further boosts the anti-PD-1 therapy with complete metastasis remission in a late-stage melanoma mouse model by fortifying the NKs and CTLs responses. An in vivo functional study demonstrates that the improved therapeutic efficacy is CD8-, IFN-γ-, NKG2D- and NK-dependent. Furthermore, DTIC/Epacasome prevent tumour recurrence and extend mouse survival when combined with anti-PD-1 in a clinically relevant post-surgical melanoma model.

## Results
### Building and characterisations of the Epacasome
The synthesis of the SM-EPA conjugate is delineated in Fig. 1a. The hydroxyl group of SM was reacted with succinic anhydride to produce the SM-COOH, which was subsequently conjugated to the oxime moiety of EPA to render the SM-EPA prodrug bridged by an oxime-ester bond that is highly sensitive to the hydrolase in tumours[42,43]. By attacking the hydroxidenucleophiles at the electrophilic C of the ester C = O, the π bond was broken and generated the tetrahedral intermediate. Following the intermediate collapse, the C = O bond reformed with the release of EPA. Several condensing agents (e.g.,

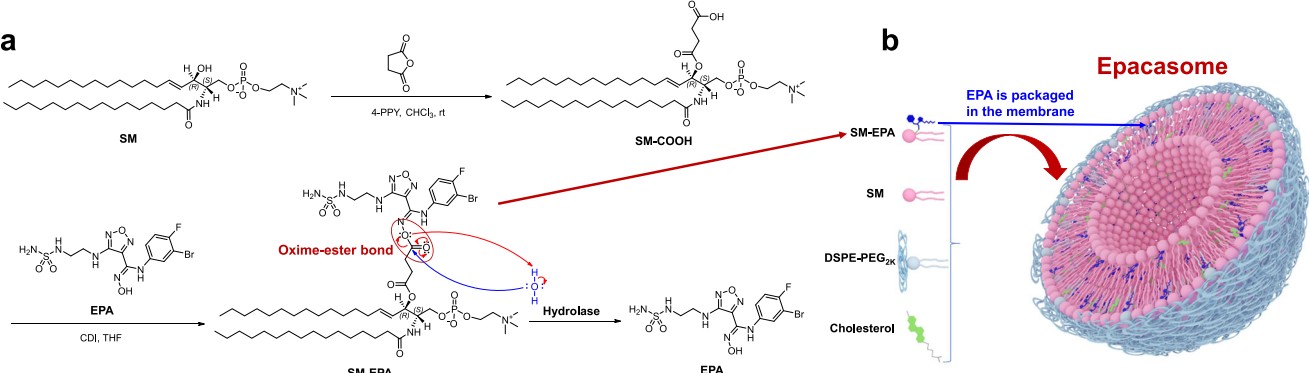

**Fig. 1 | Development of SM-derived EPA liposomal nanovesicles (Epacasome). a** Synthesis of SM-EPA bridged by an oxime-ester bond. **b** Schematic depicting the self-assembly of SM-EPA into Epacasome.

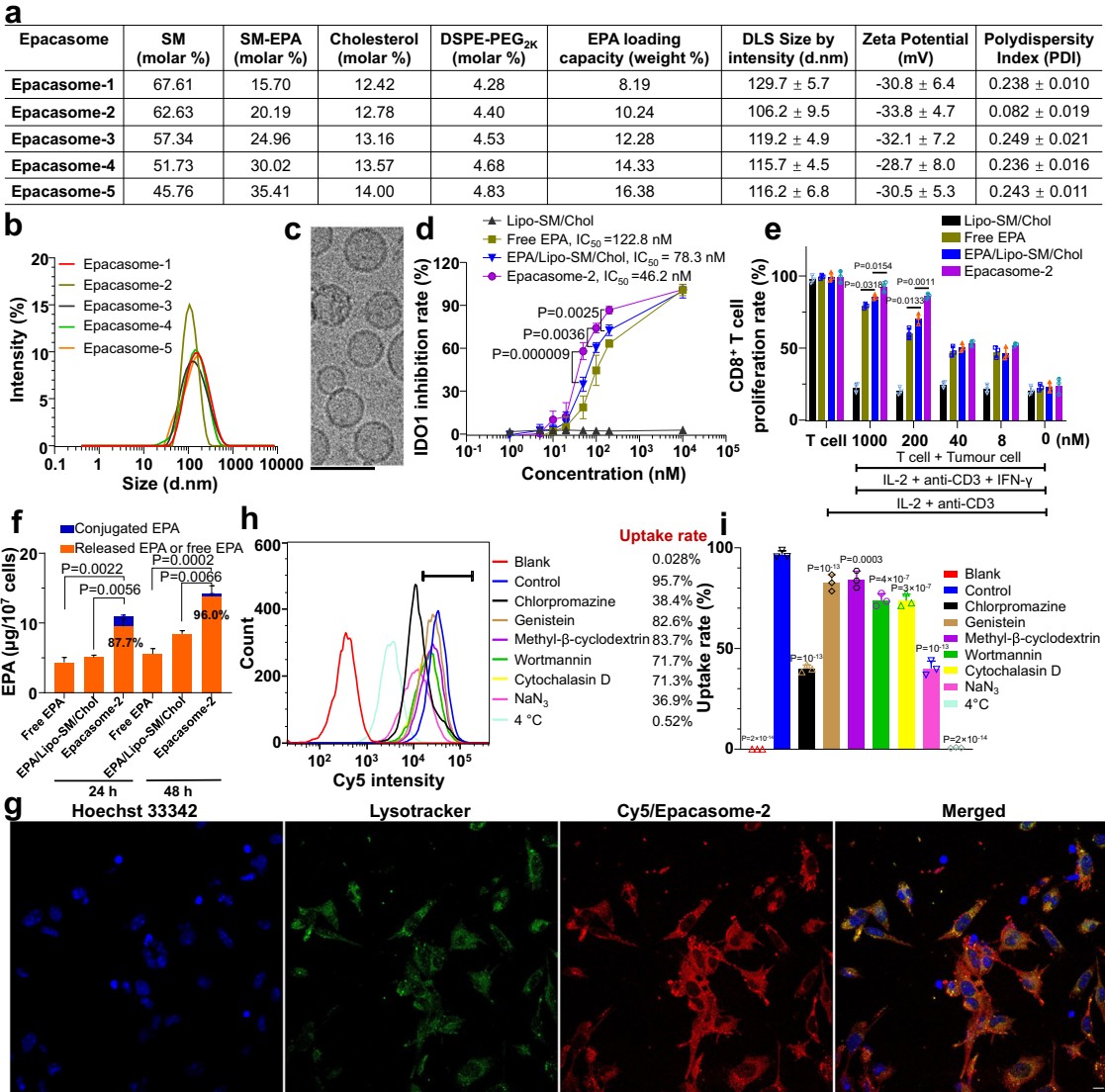

**a**

| Epacasome | SM (molar %) | SM-EPA (molar %) | Cholesterol (molar %) | DSPE-PEG$_{2K}$ (molar %) | EPA loading capacity (weight %) | DLS Size by intensity (d.nm) | Zeta Potential (mV) | Polydispersity Index (PDI) |
|---|---|---|---|---|---|---|---|---|
| Epacasome-1 | 67.61 | 15.70 | 12.42 | 4.28 | 8.19 | 129.7 ± 5.7 | -30.8 ± 6.4 | 0.238 ± 0.010 |
| Epacasome-2 | 62.63 | 20.19 | 12.78 | 4.40 | 10.24 | 106.2 ± 9.5 | -33.8 ± 4.7 | 0.082 ± 0.019 |
| Epacasome-3 | 57.34 | 24.96 | 13.16 | 4.53 | 12.28 | 119.2 ± 4.9 | -32.1 ± 7.2 | 0.249 ± 0.021 |
| Epacasome-4 | 51.73 | 30.02 | 13.57 | 4.68 | 14.33 | 115.7 ± 4.5 | -28.7 ± 8.0 | 0.236 ± 0.016 |
| Epacasome-5 | 45.76 | 35.41 | 14.00 | 4.83 | 16.38 | 116.2 ± 6.8 | -30.5 ± 5.3 | 0.243 ± 0.011 |

**Fig. 2 | Epacasome strengthened IDO1 inhibition and enhanced cellular uptake mainly via clathrin-mediated endocytosis. a** A table illustrating the physicochemical properties of Epacasome composed of SM/SM-EPA/Cholesterol/DSPE-PEG$_{2K}$ with various molar ratios. d.nm diameter values in nanometres, DLS dynamic light scattering. **b** DLS size distribution by intensity. **c** Cryo-electron microscopy (cryo-EM) of Epacasome-2 ($n = 3$ independent experiments, similar results were observed). Scale bar, 100 nm. **d** IDO1 enzymatic inhibitory activity via measuring the Kyn in supernatants in HeLa cells treated with IFN-γ along with free EPA, Lipo-SM/Chol, EPA/Lipo-SM/Chol and Epacasome-2 at equivalent (eq.) EPA concentration (0, 1, 5, 10, 20, 50, 100, 200 and 10,000 nM.) for 48 h. **e** T-cell proliferation by co-culture[46,65]. B16−F10 cells were stimulated by IFN-γ to induce IDO1 expression, then treated by Mitomycin C prior to mixing with splenocytes. Free EPA, Lipo-SM/Chol, EPA/Lipo-SM/Chol and Epacasome-2 were added to co-culture cells at eq. dose of EPA (0, 8, 40, 200 and 1000 nM). To evaluate T-cell proliferation, anti-CD3 and IL-2 were added to co-cultures. Three days later, CD8$^+$ T-cell proliferation was assessed by FACS analysis. **f** Cellular uptake levels and EPA release of Epacasome-2 in B16−F10 cells after 24- and 48-h incubation measured by HPLC, respectively (Supplementary Fig. 3). The release ratios of EPA from Epacasome-2 were presented as percentage values in the figures. **g** Representative confocal laser scanning microscopy (CLSM) for visualising the intracellular trafficking of Cy5/Epacasome-2 in B16−F10 cells after incubating with Cy5/Epacasome-2 for 2 h. Hoechst 33342 (blue) and Lysotracker (green) were used to stain cell nuclei and lysosome, respectively ($n = 3$ independent experiments, similar results were observed). Scale bar, 10 μm. **h, i** Mechanistic investigation of cellular internalisation. B16−F10 cells were treated with various endocytotic inhibitors first for 0.5 h then incubated with Epacasome-2 labelled by DSPE-Cy5 for another 2 h. **h** Representative Cy5 intensity histogram by flow cytometry. **i** Cellular uptake by cell count percentage analysis. Data in (**a**, right portion, **d**–**f**, **i**) are expressed as mean ± s.d. ($n = 3$ biologically independent samples). Statistical significance was determined by one-way ANOVA followed by Tukey's multiple comparisons test. Source data are provided as a Source Data file.

DCC, EDCI, HATU, etc.) were initially employed for the conjugation but only met with a limited success rate. Finally, we decided on CDI that imparted the optimal synthetic yield for SM-EPA. The oxime-ester bond is theoretically more labile than regular ester under high hydrolase levels in the tumour microenvironment (TME) due to the p-π conjugation in the oxime[43]. The chemical structure of the conjugate was confirmed by $^1$H-NMR, $^{13}$C-NMR, and high-resolution mass spectrometry (HRMS) (Supplementary Fig. 1). Details of synthetic methods are shown in the method sections. To prepare the Epacasome, five

different molar ratios of SM/SM-EPA/Cholesterol (Chol)/DSPE-PEG$_{2K}$ were used with EPA drug loading capacity (DLC) from 8.19 to 16.13% (Fig. 2a). Upon aqueous suspensions, the amphiphilic SM-EPA self-assembled into nanovesicles with size ~106−130 nm, among which the Epacasome-2 displayed smaller size, optimal uniformity (smaller PDI), as well as better stability (Fig. 2a−c and Supplementary Fig. 5a, b). Therefore, Epacasome-2 was used for the subsequent studies. We have used liposome (Lipo) consisting of SM/Chol/PEG (Lipo-SM/Chol, at the same lipid ratio as Epacasome-2) to physically load EPA. In addition,

physically loading EPA into the liposomes composed of DOPE/Chol/DSPE-PEG$_{2K}$ (Lipo-DOPE/Chol) and HSPC/Chol/DSPE-PEG$_{2K}$ (Lipo-HSPC/Chol) have also been investigated according to the published literature[44,45]. Our results showed that the EPA loading capacity was around 0.23–1.27% through physically encapsulating EPA into various liposomes, which were significantly increased in Epacasome (8.19–16.38%); and physically loaded EPA/Lipo also had poorer stability than Epacasome (Fig. 2a and Supplementary Fig. 5). Thus, the markedly improved drug loading capacity (DLC) and stability justified the use of the SM-EPA conjugation strategy.

The ability of Epacasome-2 to inhibit the IDO1 enzymatic activity was evaluated by measuring the conversion of Trp to Kyn using a colorimetric assay[46]. Free EPA showed a concentration-dependent reduction of Kyn, demonstrating its potent IDO1 inhibition. Interestingly, Ecapasome-2 outperformed the free EPA and EPA/Lipo-SM-Chol in suppressing IDO1 as reflected by the decreased IC$_{50}$ (46.2 vs 122.8 nM) (Fig. 2d). Similar results were also observed in 4T1 cancer cells (Supplementary Fig. 9d). We then queried if the enhanced IDO1 inhibition of Epacasome-2 would lead to better reversion of T-cell suppression mediated by IDO1-expressing B16−F10 melanoma cells[39]. To test this hypothesis, we employed the co-culture assay by mixing the splenocytes and B16−F10 cells that were treated with free EPA and Eapacsome-2 along with IFN-γ, anti-CD3 and IL-2. Via co-culture assay, we proved that Epacasome-2 markedly increased the number of CD8$^+$ T cells compared to that of free EPA and EPA/Lipo-SM/Chol (Fig. 2e and Supplementary Fig. 8), demonstrating the stronger potential of Epacasome-2 to induce T-cell proliferation. To decipher why Epacasome-2 had better effects, we assessed the cellular uptake rate and EPA release in B16−F10 cells. Our results showed that Ecapasome-2 was ingested into cells at a rate 2–3-fold (2.51 and 2.53-fold at 24 and 48 h, respectively) higher than free EPA with rapid and efficient drug release inside cells (Fig. 2f and Supplementary Fig. 9a–c), supporting the enhanced IDO1 inhibition and T-cell proliferation observed (Fig. 2d, e). To rule out the possible immunostimulatory properties from SM or its metabolism, we used SM to prepare the empty liposome, Lipo-SM/Chol with the same lipid ratio as Epacasome-2 (Supplementary Fig. 5c–f). We demonstrated that Lipo-SM/Chol had no effect on IDO1 inhibition and did not render any noticeable benefits for CD8$^+$ T-cell proliferation (Fig. 2d, e).

To investigate the uptake mechanism, we labelled Epacasome-2 with a Cy5 dye via doping DSPE-Cy5 into the lipid bilayer. After a serial titration, we determined that at 0.2% (w/w) of DSPE-Cy5, Cy5/Epacasome-2 had the closest size, zeta potential and PDI as Epacasome-2 without dye (Supplementary Fig. 11). Our confocal imaging demonstrated that the punctate distribution of Cy5/Epacasome-2 was well co-localised with the lysosome, suggesting that Cy5/Epacasome-2 was taken up by cells via classic endocytosis (Fig. 2g). To delve deeper into the mechanism for cellular uptake, cells were pretreated with different endocytotic inhibitors. We found that the uptake of Cy5/Epacasome-2 was completely or substantially blocked by 4 °C or NaN$_3$, revealing that its internalisation was energy-dependent. While the uptake rate was inhibited by the caveolar inhibitors (genistein and methyl-β-cyclodextrin) or the macropinocytosis inhibitors (wortmannin and cytochalasin D), the clathrin-mediated endocytosis inhibitor (chlorpromazine) blocked the uptake the most. These data elucidated that Epacasome-2 was taken up by cells mainly through energy-dependent clathrin-mediated endocytosis (Fig. 2g–i).

**Blood kinetics, tissue distribution and deep tumour infiltration**
The pharmacokinetics and biodistributions of Epacasome-2 were examined in B16−F10 tumour-bearing mice. Free EPA presented rapid blood clearance kinetics with a $T_{1/2} = 0.15$ h, consistent with the literature[37], while Epacasome-2 substantially prolonged the circulatory $T_{1/2}$ from ~0.73 h in EPA/Lipo-SM/Chol to ~4.72 h (Fig. 3a, d). The improved pharmacokinetics enabled more passive targeting of Epacasome-2 (46.3 and 2.6-fold more than free EPA and EPA/Lipo-SM/Chol, respectively) to tumours based on enhanced permeability and retention effect (Fig. 3b)[47,48]. Upon reaching the tumours, within 24 h, 86.1% of EPA was liberated from the Epacasome-2 (Fig. 3c). To achieve optimal anticancer activity, the drug must effectively extravasate from the vasculature, and then penetrate and infiltrate tumours deeply and uniformly. To decipher if Epacasome-2 can do so, we labelled it with DSPE-Cy5 as established in Fig. 2 and evaluated the tumours using immunofluorescence confocal imaging (Fig. 3e). We proved that Cy5/Epacasome-2 (red signal) migrated farther away from the blood vessels (platelet endothelial cell adhesion molecule 1 (PECAM1), green signal) and distributed throughout the tumour sections better than EPA/Lipo-SM/Chol (Fig. 3e). Taken together, these findings corroborated that Epacasome-2 remained stable during circulation and provide superior tumour delivery potential, while facilitating timely release of EPA with deep and uniform tumour penetration and distribution, which are pivotal properties for in vivo EPA-mediated IDO1 inhibitory therapeutic delivery.

**Epacasome boasts excellent safety and potentiates PD-1 blockade in melanoma models**
The therapeutic efficacy of Epacasome-2 vs free EPA (either by oral (p.o.) or intravenous (i.v.) administration) was investigated in mice bearing early stage B16−F10 tumour (~100 mm$^3$) (Fig. 4). Mice received free EPA (p.o. or i.v.) or Epacasome-2 (i.v.) on day 8, 10, 12 and 14 at eq. of 41 mg EPA/kg alone (converted from clinic dose of EPA)[14] or in combination with intraperitoneal (i.p.) injection of anti-PD-1 monoclonal antibody (α-PD-1, 100 μg per mouse) every 3 day on three occasions[14]. The tumours from the vehicle control group grew uncontrollably, indicating the aggressive nature of this tumour type. Free EPA (p.o. or i.v.) or α-PD-1 alone were able to reduce the tumour growth modestly, however, which was outperformed by Epacasome-2 (Fig. 4b, c and Supplementary Fig. 12a). EPA enhanced the efficacy of α-PD-1 (in accordance with literature)[11], particularly in the form of Epacasome-2 as demonstrated by the delayed tumour development and prolonged mouse survival (Fig. 4b–d and Supplementary Fig. 12a). Systematic analysis of the median survival time (MST), time to reach endpoint (TTE), tumour growth delay (TGD), and increased live span (ILS) corroborated the therapeutic potential of Epacasome-2 + α-PD-1 (Supplementary Table 1). Moreover, Epacasome-2 beat EPA/Lipo-SM/Chol, EPA/Lipo-DOPE/Chol and EPA/Lipo-HSPC/Chol on anticancer efficacy and immune responses (Supplementary Figs. 14 and 15). Taken together, these results justified the use of Epacasome-2 to improve the translational potential of EPA. Furthermore, compared to vehicle control, empty liposome Lipo-SM/Chol exerted negligible inhibition on the tumour growth in B16−F10 tumour model and had no significant effects on the Trp/Kyn ratio in both plasma and tumours as well as the antitumour immune responses (Supplementary Fig. 14). These findings demonstrated that the observed antitumour effects of Epacasome-2 were not related to the immunomodulatory properties of SM. The fortified antimelanoma activity of Epacasome-2 could be due to the improved pharmacokinetics, and the bolstered tumour accumulation and penetration (Fig. 3). Of note, under the current dosing regimen, no overt systemic toxicities were observed in healthy mice for either free EPA or Epacasome-2, or their respective combination regimen, which was evidenced by stability of the comprehensive blood chemistry and haematological cell counts such as leukocytes, erythrocyte, and thrombocytes, as well as the histology study of liver, spleen, kidney, muscle and intestinal mucosa samples (Supplementary Fig. 13), indicative of a superior safety profile for Epacasome-2. To probe the in vivo IDO1 effects and antitumour immune responses induced by Epacasome-2, an independent animal study was performed as Fig. 4a. We demonstrated that the ratios of Trp/Kyn in both plasma and tumours were markedly increased in the EPA group

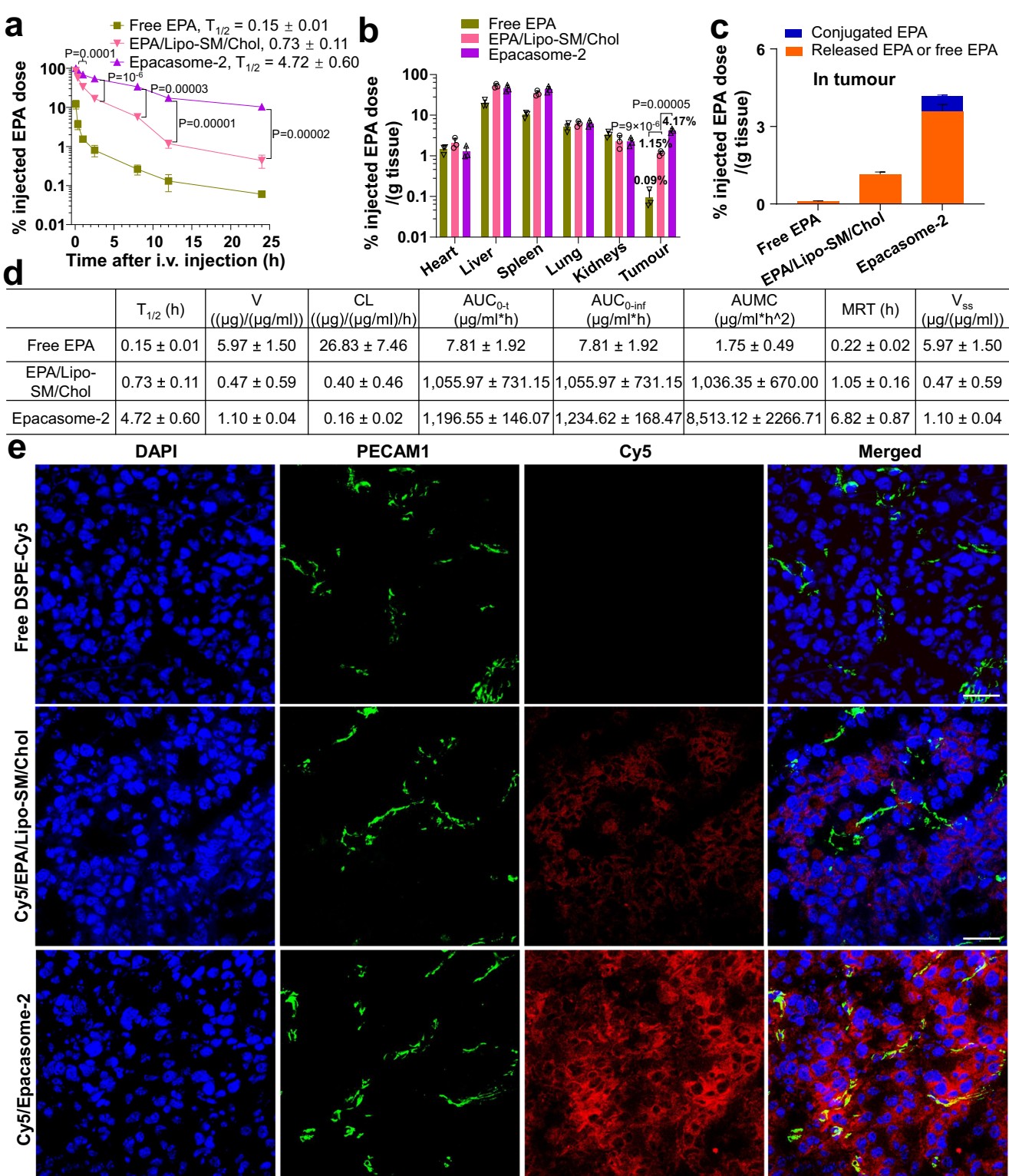

**Fig. 3 | Improved pharmacokinetics and tumour delivery with efficient EPA release and deep penetration in tumours. a–d** Blood kinetics (**a**), biodistribution (**b**) and EPA intratumoural release (**c**) at 24 h in mice from (**a**) and various pharmacokinetic parameters (**d**) in B16–F10 tumour-bearing C57BL/6 mice (*n* = 3 mice; tumour: ~400 mm³) following a single i.v. administration of free EPA, EPA/Lipo-SM/Chol and Epacasome-2 at eq. 10 mg EPA/kg. The per cent injected doses in Epacasome represent the released EPA and SM-conjugated EPA. Drug contents in plasma and major tissues were measured by HPLC (Supplementary Fig. 3). **e** Unravelling the potential of Epacasome-2 for extravasating and penetrating the tumours post an i.v. injection into mice bearing subcutaneous B16–F10 tumours (*n* = 3 mice, tumours, ~400 mm³). Twenty-four hours after injecting Cy5/Epacasome (red), CLSM of sections of B16–F10 tumours was conducted. Cell nuclei were stained by DAPI (blue). Blood vessels were stained with PECAM1 antibody followed by Alexa Fluor 488 secondary antibody staining (green). Scale bars, 50 μm. Data in (**a–d**) are expressed as mean ± s.d. (*n* = 3 mice). Statistical significance was determined by one-way ANOVA followed by Tukey's multiple comparisons test. Source data are provided as a Source Data file.

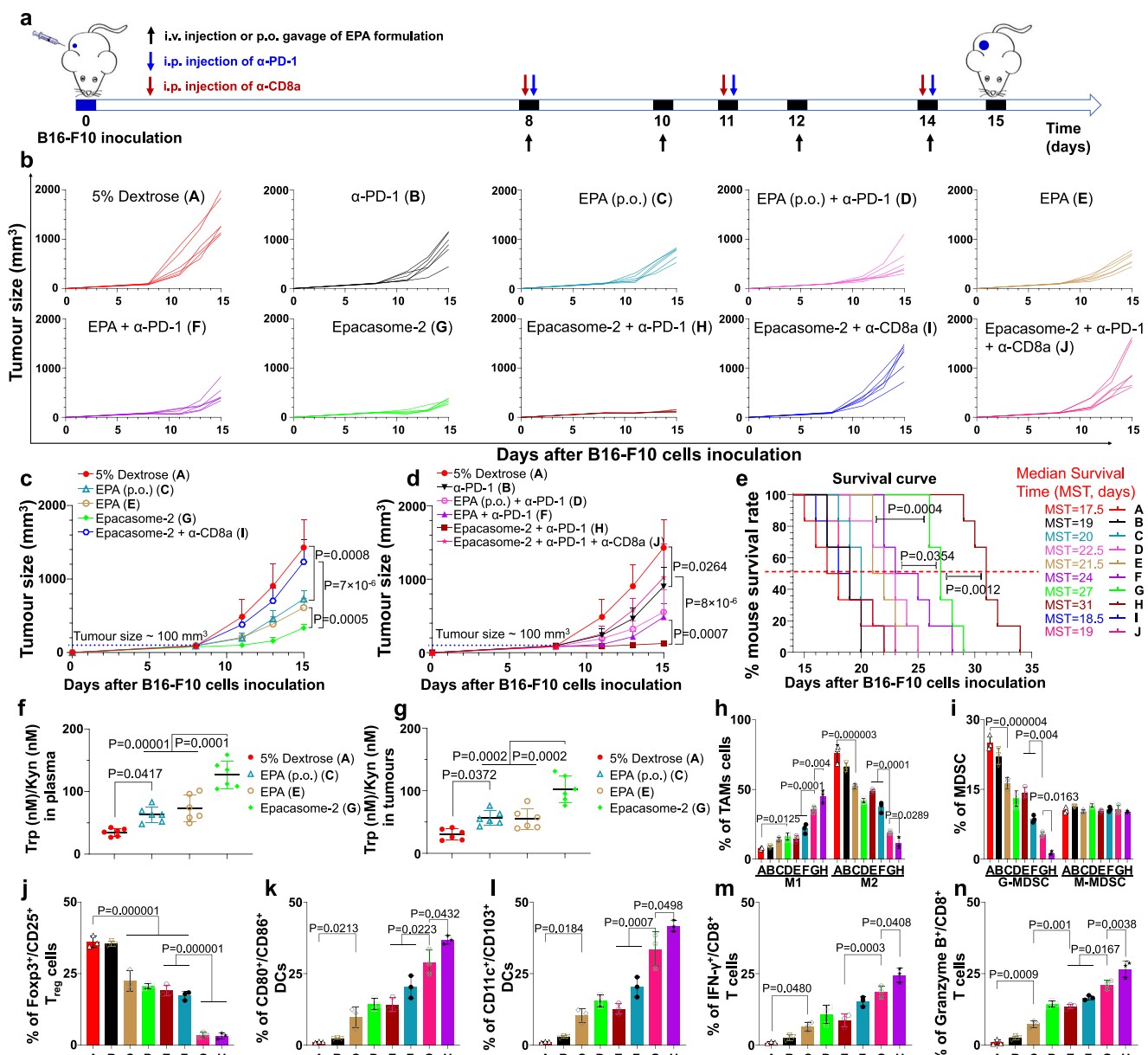

**Fig. 4 | Epacasome-2 potentiated PD-1 blockade in B16–F10 tumour mice via inhibiting IDO1 and eliciting potent CTL responses. a** Drug administration timeline scheme in subcutaneous (s.c.) B16–F10 tumour model ($n = 6$ mice, tumours: ~100 mm³). Mice were intravenously injected with Epacasome-2 or free EPA (p.o. or i.v.) at eq. 41 mg EPA/kg on days 8, 10, 12 and 14 alone or combined with i.p. α-PD-1 (BioXCell, clone RMP1-14, 100 μg per mouse per 3 day for 3 times) from day 8[39, 69]. **b** Individual tumour growth curves. **c, d** Average tumour size growth curves of EPA formulation alone (**c**) or combined with α-PD-1 (**d**). **e** Kaplan–Meier survival curves. **f, g** An independent efficacy study in B16–F10 tumour mice as (**a**). **f, g** The ratio of Trp (nM)/kyn (nM) concentration in plasma (**f**) and tumours (**g**),

($n = 6$ mice). **h–n** Quantification analysis of intratumoural CD45⁺/CD11b⁺/F4/80⁺/CD206⁻ M1 and CD45⁺/CD11b⁺/F4/80⁺/CD206⁺ M2 TAMs cells (**h**)[46, 49], CD45⁺/CD11b⁺/Gr-1⁺ MDSC cells (**i**), CD3⁺/CD4⁺/Foxp3⁺/CD25⁺ Tregs (**j**), CD45⁺/CD11c⁺/CD80⁺/CD86⁺ DCs (**k**), CD45⁺/CD11c⁺/CD103⁺ DCs (**l**), Granzyme B⁺ (**m**) or IFN-γ⁺/CD3⁺/CD8⁺ (**n**) T cells from three random chosen tumours by flow cytometry. The representative flow cytometric plots and gating strategies were placed in Supplementary Figs. 12b, 20 and 22. Data in (**c**, **d**, **f–n**) are expressed as mean ± s.d. Statistical significance was determined by one-way ANOVA followed by Tukey's multiple comparisons test; survival curves were compared using the log-rank Mantel–Cox test. Source data are provided as a Source Data file.

(Fig. 4f, g), which were further boosted by Epacasome-2, suggesting the potent suppression of IDO1 activity in vivo. Using flow cytometry, we substantiated that free EPA was capable of eliciting CTL responses via activation of the DCs (increased CD80⁺/CD86⁺) and CD11c⁺/CD103⁺, upregulation of IFN-γ⁺ or Granzyme B⁺ CD8⁺ T cells, and dampening of Foxp3⁺ Tregs and granulocytes MDSC (G-MDSC) as well as repolarization of the TAM (enhanced M1/M2 ratios) immunosuppressive effects in tumours (Fig. 4h–n)[46,49]. These anti-melanoma immune effects were notably enhanced in Epacasome-2 compared to free EPA, especially when combined with α-PD-1 therapy (Fig. 4h–n). Upon systemically knocking down the CD8a, the

antitumour efficacy and immune effects of Epacasome-2 were almost completely abrogated (Fig. 4b–e). These data confirmed that Epacasome-2 can serve as a more powerful immunostimulatory agent to reverse tumour immunosuppression for better antitumour CTL effects by bolstering the IDO1 inhibition.

## Epacasome-2 synchronised therapeutic delivery of DTIC and EPA for improved immunotherapy in advanced melanoma models

DTIC is used for the treatment of metastatic melanoma. It has been reported that DTIC can elicit natural killer (NK) and CD8⁺ T-cell

immune responses by upregulating NKG2D ligands on melanoma tumour cells[50]. To elucidate the optimal drug ratio, we have systemically screened and evaluated the combination ratios of DTIC and SM-EPA based on the synergistic effect in B16−F10 cells (Fig. 5a−c). The combination index (CI) revealed that DTIC/SM-EPA with the molar ratio of 3/1 and 4/1 yielded the lowest CI values (0.40 and 0.41, respectively), which suggested the potent synergy in inhibiting B16−F10 cell proliferation. Considering the maximum tolerated dose for DTIC, the fixed amount of EPA on the bilayer of Epacasome-2 as well as the EPA dose for therapeutic efficacy studies converted from the clinical application[14, 51], the molar ratio of 3.66/1 (DTIC/SM-EPA, 75 mg DTIC/kg and 41 mg EPA/kg) were considered to be the most appropriate and were used for the efficacy studies. We also demonstrated that the CI of 3.66/1 (DTIC/SM-EPA) is 0.40, confirming this ratio is optimal. To investigate whether DTIC can synergise Epacasome-2 to further enhance the PD-1 blockade for improved anti-melanoma immunotherapy, we proposed to co-deliver DTIC using Epacasome-2. Owing to the poor solubility of free DTIC, its direct encapsulation into Epacasome-2 only had limited success (DLC: 0.24%). To facilitate DTIC loading, it was converted into a salt form (DTIC·HCl) by generating hydrochloride (Fig. 5d). This strategy resulted in much higher DLCs (up to 40.13%) in DTIC/Epacasome-2 following the direct film hydration approach with a uniform size distribution (Fig. 5e and Supplementary Fig. 16). We used ratio-2 that carries 15.77% DTIC for in vivo studies because of the maximum tolerated dose for DTIC and the fixed amount of EPA on the bilayer of Epacasome-2 (Fig. 5g)[51]. We found that free DTIC had a very short circulation half-life (~0.65 h), and DTIC loaded in liposome consisting of 82.82% SM, 12.78% Chol, and 4.40% DSPE-PEG$_{2K}$ can improve the pharmacokinetics and tumour delivery (Fig. 5i−k), demonstrating the advantages of using a nanocarrier to deliver DTIC. However, DTIC/Epacasome-2 was able to further extend the circulation time by increasing the half-life of DTIC from ~1.04 h in DTIC/Lipo-SM/Chol to ~4.44 h with markedly improved area under curve (AUC) and mean residence time (MRT) (Fig. 5k) and delivered twofold more DTIC to tumours (Fig. 5j). Compared to co-administration of DTIC/Lipo-SM/Chol and Epacasome-2, co-delivery DTIC/Epacasome-2 not only improved the PK and tumour accumulation, but also entailed synchronised therapeutic delivery for both DTIC and EPA (Fig. 5l−n and Supplementary Table 2).

In a late-stage metastatic B16−F10-Luc2 melanoma model, we showed that α-PD-1 had no control over tumour growth or mitigating the widespread metastasis, demonstrating how this advanced melanoma model had poor response to PD-1 blockade therapy (Fig. 6a, b). Free DTIC showed modest tumour reduction, and this effect was further improved when DTIC was formulated in liposome (DTIC/Lipo-SM/Chol). Epacasome-2 also had noticeable tumour inhibition and metastasis attenuation (Fig. 6b−f). To elucidate if the co-delivery approach works better than co-administration, we intravenously injected DTIC/Lipo-SM/Chol + Epacasome-2 together into tumour bearing mice and compared this dosing regimen to co-delivery DTIC/Epacasome-2. Our results proved that co-delivering DTIC and EPA in DTIC/Epacasome-2 yielded a significantly stronger tumour inhibitory effect and better prevention of tumour metastasis over co-administrating DTIC/Lipo-SM/Chol + Epacasome-2; Our results also showed that the co-delivery further strengthened the activity of anti-PD-1 blockade therapy to shrink tumours to ~1/4 their starting size and exhaustively eradicated metastasis in 6 out of 6 mice (Fig. 6b−f). DTIC/Epacasome-2 prolonged survival rate compared to DTIC/Lipo-SM/Chol + Epacasome-2, particularly when combined with α-PD-1. (Supplementary Fig. 18d). By comparing the co-administration of DTIC/Lipo-SM/Chol and Epacasome-2 or monotherapy alone, we elucidated that DTIC/Epacasome-2 induced noticeably higher tumour-infiltrating levels of IL-2, especially in combination with α-PD-1 (Fig. 6g). To confirm if DTIC can upregulate the NKG2D ligands (NKG2D$^L$) on tumour cells, on day 18, the tumours from Fig. 6a were subject to the flow cytometric analysis for Rae-1 and Mult-1 NKG2D$^L$. We

showed that DTIC not only markedly upregulated Rae-1 and Mult-1 but also upregulated MHC-I expression on tumour cells, particularly in the form of DTIC/Ecapasome-2 (Fig. 6h−m). These findings corroborated the immunogenic potential of DTIC[50, 52]. To elucidate the underlying mechanisms associated with the improved therapeutic outcomes of DTIC/Epacasome-2, we then systemically investigated the NKG2D$^+$(receptor), CD69$^+$, IFN-γ$^+$, Granzyme B$^+$ or perforin$^+$ NK and CD8$^+$ T cells respectively, in tumours using flow cytometry. We demonstrated that DTIC/Epacasome-2 elicited markedly higher tumour-infiltrating levels of NKG2D (receptor)$^+$/NK$^+$ cells, CD69$^+$/NK$^+$ cells, IFN-γ$^+$/NK$^+$ cells, Perforin$^+$/NK$^+$ cells, and Granzyme B$^+$/NK$^+$ cells, as well as NKG2D (receptor)$^+$/CD8$^+$ T cells, CD69$^+$/CD8$^+$ T cells, IFN-γ$^+$/CD8$^+$ T cells, Perforin$^+$/CD8$^+$ T cells and Granzyme B$^+$/CD8$^+$ T cells, in comparison to the co-administration of DTIC/Lipo-SM/Chol and Epacasome-2 or monotherapy alone. We additionally demonstrated that the combination with α-PD-1 further reinforced these anti-melanoma immune responses (Fig. 6n and Supplementary Fig. 19). Interestingly, the levels of PD-1$^+$/CD8$^+$ exhausted T cells were increased in groups containing DTIC/Lipo, especially in the DTIC/Epacasome-2. However, no significant differences of LAG-3$^+$/CD8$^+$ and Tim-3$^+$/CD8$^+$ T cells were observed after the treatments. These data justified the combination with α-PD-1 therapy to further enhance therapeutic outcome.

To validate that the improved antitumour effects were indeed correlated with the enhanced host immune responses, CD8α, NK1.1, IFN-γ, and NKG2D were systemically depleted using anti-CD8α, NK1.1, IFN-γ, or NKG2D antibodies[39, 50] (Fig. 7). Evaluation through intratumoural IHC confirmed the depletion of the corresponding immune cell subsets (Supplementary Fig. 24c). We discerned that elimination of CD8α and IFN-γ abolished most of the antitumour activity of DTIC/Epacasome-2. Depletion of NK markedly reduced the efficacy to a level that was less effective than Epacasome-2 alone, whereas eliminating NKG2D led to anticancer activity compared to Epacasome-2 alone. These discoveries underpinned that DTIC/Epacasome-2 antitumour effects were indispensably dependent on CD8$^+$ T cells and IFN-γ and were also partially dependent on NKs and NKG2D.

## Combination with α-PD-1 prevents post-surgical melanoma recurrence

While surgery can remove the gross tumour mass, microscopic residual residue tumour cells can sometimes lead to more aggressive recurrences[53]. To determine whether DTIC/Epacasome-2 can overcome tumour relapse following surgical resection of late-stage melanoma, we established a clinically relevant post-surgical melanoma model (Fig. 8a). Thirteen days after the B16−F10-Luc2 cells were inoculated, the primary tumours (~300 mm³) were surgically excised, while intentionally leaving ~1% of original tumour mass as residual microtumours[54]. One day later, mice received various corresponding treatments (Fig. 8a, b). Our data show that after surgical excision, the tumours grew back rapidly in vehicle control mice. α-PD-1 blockade only marginally mitigated the tumour relapse at the surgical site without offering significant mouse survival benefit (Fig. 8b−e). Noticeably, in stark contrast to co-administration of DTIC/Lipo-SM/Chol and Epacasome-2, DTIC/Epacasome-2 was far more effective in preventing tumour recurrence with longer survival time. DTIC/Epacasome-2 in combination with PD-1 blockade led to complete tumour control while vehicle control tumours reached the endpoint on day 22. Finally, DTIC/Epacasome-2 drastically prolonged the mouse survival rate (Fig. 8).

## Discussion

The failure of the combinatorial therapy of EPA and PD-1 inhibitor (Keytruda) in a phase III clinical trial in melanoma patients has halted the field of IDO-based therapeutic development and clinical translation. The dosing regimen and broad patient samples (only patients with T-cell-inflamed tumours react to the immunotherapy) have been

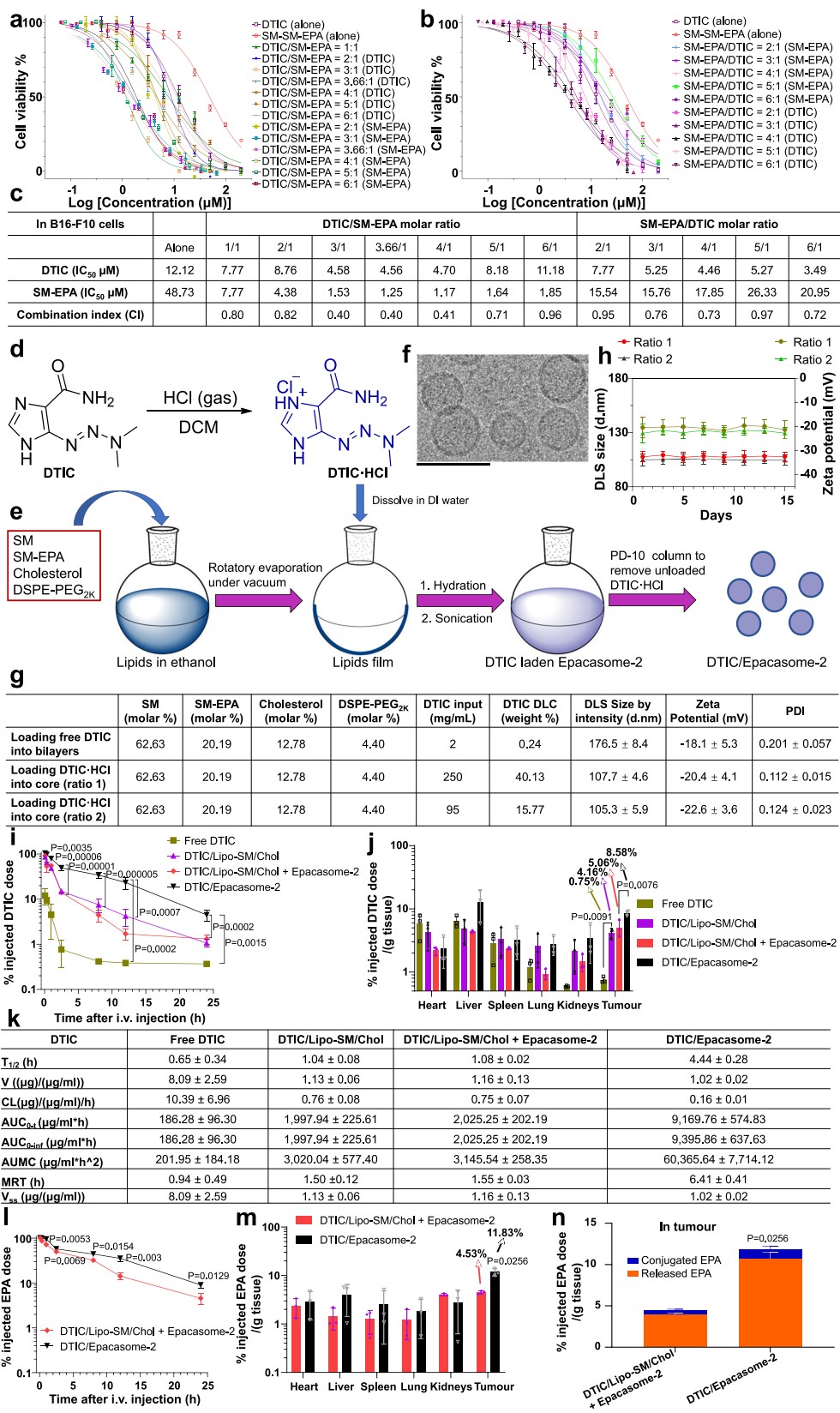

| In B16-F10 cells | | DTIC/SM-EPA molar ratio | | | | | | | SM-EPA/DTIC molar ratio | | | | |
|---|---|---|---|---|---|---|---|---|---|---|---|---|---|
| | Alone | 1/1 | 2/1 | 3/1 | 3.66/1 | 4/1 | 5/1 | 6/1 | 2/1 | 3/1 | 4/1 | 5/1 | 6/1 |
| DTIC (IC$_{50}$ μM) | 12.12 | 7.77 | 8.76 | 4.58 | 4.56 | 4.70 | 8.18 | 11.18 | 7.77 | 5.25 | 4.46 | 5.27 | 3.49 |
| SM-EPA (IC$_{50}$ μM) | 48.73 | 7.77 | 4.38 | 1.53 | 1.25 | 1.17 | 1.64 | 1.85 | 15.54 | 15.76 | 17.85 | 26.33 | 20.95 |
| Combination index (CI) | | 0.80 | 0.82 | 0.40 | 0.40 | 0.41 | 0.71 | 0.96 | 0.95 | 0.76 | 0.73 | 0.97 | 0.72 |

| | SM (molar %) | SM-EPA (molar %) | Cholesterol (molar %) | DSPE-PEG$_{2K}$ (molar %) | DTIC input (mg/mL) | DTIC DLC (weight %) | DLS Size by intensity (d.nm) | Zeta Potential (mV) | PDI |
|---|---|---|---|---|---|---|---|---|---|
| Loading free DTIC into bilayers | 62.63 | 20.19 | 12.78 | 4.40 | 2 | 0.24 | 176.5 ± 8.4 | -18.1 ± 5.3 | 0.201 ± 0.057 |
| Loading DTIC·HCl into core (ratio 1) | 62.63 | 20.19 | 12.78 | 4.40 | 250 | 40.13 | 107.7 ± 4.6 | -20.4 ± 4.1 | 0.112 ± 0.015 |
| Loading DTIC·HCl into core (ratio 2) | 62.63 | 20.19 | 12.78 | 4.40 | 95 | 15.77 | 105.3 ± 5.9 | -22.6 ± 3.6 | 0.124 ± 0.023 |

| DTIC | Free DTIC | DTIC/Lipo-SM/Chol | DTIC/Lipo-SM/Chol + Epacasome-2 | DTIC/Epacasome-2 |
|---|---|---|---|---|
| T$_{1/2}$ (h) | 0.65 ± 0.34 | 1.04 ± 0.08 | 1.08 ± 0.02 | 4.44 ± 0.28 |
| V ((μg)/(μg/ml)) | 8.09 ± 2.59 | 1.13 ± 0.06 | 1.16 ± 0.13 | 1.02 ± 0.02 |
| CL(μg)/(μg/ml)/h) | 10.39 ± 6.96 | 0.76 ± 0.08 | 0.75 ± 0.07 | 0.16 ± 0.01 |
| AUC$_{0-t}$ (μg/ml*h) | 186.28 ± 96.30 | 1,997.94 ± 225.61 | 2,025.25 ± 202.19 | 9,169.76 ± 574.83 |
| AUC$_{0-inf}$ (μg/ml*h) | 186.28 ± 96.30 | 1,997.94 ± 225.61 | 2,025.25 ± 202.19 | 9,395.86 ± 637.63 |
| AUMC (μg/ml*h^2) | 201.95 ± 184.18 | 3,020.04 ± 577.40 | 3,145.54 ± 258.35 | 60,365.64 ± 7,714.12 |
| MRT (h) | 0.94 ± 0.49 | 1.50 ±0.12 | 1.55 ± 0.03 | 6.41 ± 0.41 |
| V$_{ss}$ (μg/(μg/ml)) | 8.09 ± 2.59 | 1.13 ± 0.06 | 1.16 ± 0.13 | 1.02 ± 0.02 |

thought to account for the disappointing clinical outcomes[15]. However, the pharmacokinetic profiles of EPA have not been considered and were found to be extremely poor, which could be one of the major factors leading to the clinical failure. We believe that the improved pharmacokinetics and tumour delivery efficiency achieved by Epacasome will revive its clinical translational potential.

With the central aim to rescue EPA, we developed an SM-derived EPA nanovesicle delivery platform. To do so, disulfide and carbonate ester bonds were initially applied to bridge the SM and EPA. Nevertheless, both strategies generated the oxadiazole and formation of an inactive EPA derivative through spontaneous cyclization[55]. To tackle these challenges, based on the unique chemical structures of EPA and

**Fig. 5 | Development of DTIC-co-delivered Epacasome-2. a–c** Investigation of the optimal drug ratios in B16–F10 by MTT assay. **a, b** Cells were treated at various drug combination ratios for 48 h (*n* = 3 biologically independent samples). **c** The combination index (CI) was calculated based on the IC$_{50}$ values of individual drug. **d** Synthesis of hydrochloride DTIC (DTIC·HCl) to facilitate the loading of DTIC to Epacasome-2. **e–h** Schematic to show the manufacturing process (**e**), cryo-EM (**f**, scale bar, 100 nm, triplicates were performed independently with similar results), a table delineating the physicochemical characterisations (**g**), and DLS size, zeta potential monitoring of DTIC/Epacasome-2 under 4 °C for two weeks (**h**, *n* = 3 independent experiments). **i–n** Blood kinetics (**i, l**), biodistribution (**j, m**), and EPA intratumoural release (**n**) at 24 h in mice from (**i**) and various pharmacokinetic parameters of DTIC (**k**) in B16–F10 tumour-bearing C57BL/6 mice (*n* = 3 mice; tumour: -400 mm³) following a single i.v. administration at eq. 75 mg DTIC/kg, EPA 41 mg EPA/kg. The per cent injected doses in Epacasome-2 represent the released EPA and SM-conjugated EPA. Drug contents in plasma and major tissues were measured by HPLC (Supplementary Figs. 3 and 4). DTIC/Lipo-SM/Chol consisted of 82.82% SM, 12.78% Chol, 4.40% DSPE-PEG$_{2K}$ (molar ratio) and had 15.86% DTIC drug loading capacity (DLC) (Supplementary Fig. 17). Data in (**a, b, g**, right portion), **h–n** are expressed as mean ± s.d. Statistical significance in (**i, j**) was determined by one-way ANOVA followed by Tukey's multiple comparisons test; two-tailed, unpaired Student's *t* test for (**l–n**). Source data are provided as a Source Data file.

SM, we designed a highly tumour-sensitive oxime-ester bond to covalently link EPA to the SM. This strategy circumvented the untoward cyclization and allowed EPA to be securely anchored in the lipid bilayer upon self-assembly of SM-EPA into nanovesicles (Epacasome) (Supplementary Figs. 5 and 6), protecting EPA from rapid blood clearance, drastically improving the pharmacokinetics by increasing the circulation half-life (0.15 vs 4.72 h) and area under curve (AUC, 7.81 vs -1200 μg/ml*h), reducing volume of distribution (5.97 vs 1.1 (μg)/(μg/ml)) and clearance (CL, 26.83 vs 0.16 (μg)/(μg/ml)/h) and delivering markedly more EPA into melanoma (0.09 vs 4.17%) (Figs. 1a, b, 2a–e and 3). Moreover, comparing to physically encapsulating EPA in the liposomes, including the reported EPA/Lipo-DOPE/Chol and EPA/Lipo-HSPC/Chol[44,45], Epacasome possessed significantly higher EPA DLC (8.19–16.38% vs 0.23–1.27%, Fig. 1a and Supplementary Fig. 5c). Also, Epacasome-2 markedly improved the pharmacokinetics by extending the circulation half-life (4.72 vs 0.72 h) and delivering more EPA into the tumour (4.17 vs 1.05%) with better tumour penetration efficiency as compared to physically loading EPA into liposome consisting of the SM/Chol/DSPE-PEG2K (EPA/Lipo-SM/Chol, Fig. 3). The improved pharmacokinetics and tumour distribution could be attributed to the enhanced formulation stability of Epacasome-2 (Supplementary Fig. 5a, b, e, f).

Once inside the tumours, the oxime-ester bond can be readily cleaved by the high levels of hydrolase and resulting hydrolysis, hence releasing parental EPA, while allowing Epacasome-2 to maintain a high degree of integrity during circulation (Figs. 3c and 5n and Supplementary Fig. 10). Along with the clathrin-mediated endocytosis mechanism (Fig. 2h, i), this SM-EPA prodrug strategy may avoid drug resistance by bypassing the drug efflux pump (e.g., p-glycoprotein)[56], resulting in much higher intracellular uptake and longer drug retention inside cancer cells (Fig. 2f–i). Moreover, the Epacasome-2 can efficiently extravasate from the blood vessels and penetrate deeply and uniformly inside tumours (Fig. 3e), which is crucial to achieving robust antitumour activity. These improved physicochemical properties equipped Epacasome-2 with significantly enhanced anti-melanoma activities and enabled more tumour-infiltrating CTLs, DCs, reduced immunosuppressive effects from Tregs, G-MDSCs and repolarized the macrophages through more effective targeting of the IDO1 enzyme, as well as allowed Epacasome-2 to possess better ability to potentiate the efficacy of the PD-1 inhibitor compared with free EPA (Fig. 4). In addition, Epacasome-2 performed significantly better than EPA/Lipo-SM/Chol, EPA/Lipo-DOPE/Chol and EPA/Lipo-HSPC/Chol on anti-melanoma efficacy and immune responses (Supplementary Figs. 14 and 15). Furthermore, Epacasome-2 did not cause any noticeable adverse effects in vivo (Supplementary Fig. 13), indicating its superior safety profile that is ideal for translational application.

DTIC has been proven to be immunogenic and capable of upregulating PD-1/PD-L1 expressions in melanoma[57], which justifies its combination with PD-1 blocker as high levels of immune checkpoints are associated with improved patient response to ICB[58]. To further boost the therapeutic index of Epacasome-2 plus anti-PD-1 antibody, we co-delivered DTIC using Epacasome-2. We demonstrated that it is impactful to convert the hydrophobic DTIC into DTIC·HCl because the salt form of DTIC enabled a drastically increased drug loading capacity (DLC) due to the large aqueous core in comparison to the very limited free DTIC that can be encased in the compact lipid bilayer (Fig. 5d, e, g). Interestingly, DTIC encapsulation in Epacasome-2 further prolonged the EPA circulation half-life and enabled markedly more EPA accumulation in tumours with less distribution to normal tissues (Figs. 3a–c and 5l–n). This could be ascribed to the improved retention in vivo because of the high drug-to-lipid ratio after encapsulating DTIC·HCl[59,60]. Moreover, DTIC encapsulated in Epacasome-2 markedly reduced the systemic toxicities caused by free DTIC as manifested by the comprehensive analysis of serum chemistry, leukocytes, erythrocytes, and thrombocytes (Supplementary Fig. 23). The fact that DTIC/Epacasome-2 markedly outperformed the co-administration of DTIC/Lipo-SM/Chol and Epacasome-2 on delaying tumour growth, preventing metastasis, and enhancing PD-1 blockade in late-stage metastatic melanoma corroborates the significance of the temporospatial controlled co-delivery approach (Figs. 5 and 6). Mechanistically, in addition to targeting IDO1, through in vivo functional analysis we demonstrated that our co-delivery nanotherapeutics can simultaneously activate the NKG2D-facilitated NKs and CTLs host immune responses arising from the immunogenic properties of DTIC (Figs. 6g–n and 7). Furthermore, the favourable therapeutic outcome obtained in a post-surgical melanoma model further substantiated the clinical translational potential of DTIC/Epacasome-2 (Fig. 8).

In summary, (1) IDO is widely found in various types of tumours; (2) SM is a naturally occurring phospholipid and Epacasome has good in vivo safety profiles; (3) there is therapeutic efficacy of Epacasome-based treatment regimen against clinically difficult-to-treat late-stage metastatic and post-surgical melanoma tumours; (4) the SM-conjugation nano-delivery strategy can also be applied to various other therapeutic agents containing an functional group (e.g., indoximod and NLG919, which are two widely used IDO inhibitors in clinical trials, both of which suffer severely from their hydrophobicity and poor pharmacokinetics)[61]; (5) the manufacturing of Epacasome or co-delivery Epacasome is simple and well established. Most importantly, our Epacasome nanotherapeutic platform portends significant clinical relevance and represents a promising strategy for IDO-based therapeutics against diverse cancers including melanoma.

## Methods

### Ethical statement
This research complies with all relevant ethical regulations. The animals were maintained under pathogen-free conditions and all animal experiments were approved by The University of Arizona Institutional Animal Care and Use Committee (IACUC).

### Cells culture
B16–F10 (Cat. CRL-6475™) and 4T1 (Cat. CRL-2539™) cell lines were obtained from UACC, B16–F10-Luc2 (Cat. CRL-6475-LUC2™), RAW 264.7 (Cat. TIB-71™), Panc02 (Cat. CRL-2553™) and HeLa (Cat. CRM-CCL-2™) cells were purchased from ATCC. B16–F10, B16–F10-Luc2 and RAW 264.7 cell lines were cultured in complete DMEM medium, 4T1 and Panc02 cell lines were cultured in complete RPMI1640 medium,

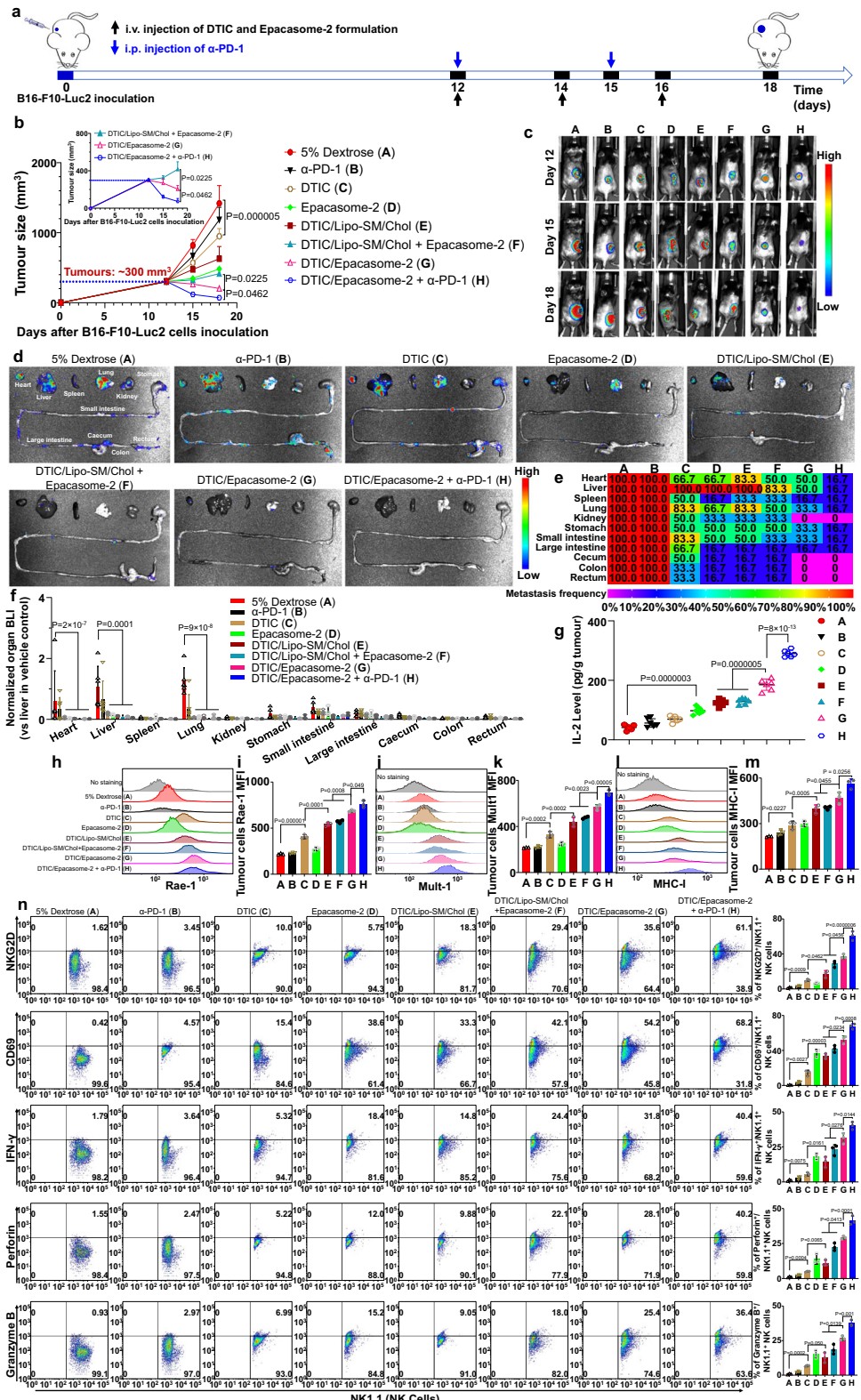

HeLa cell lines were cultured in complete MEM medium. All the cell lines were cultured in the corresponding medium containing 10% FBS, 100 U/mL penicillin, 100 µg/mL streptomycin, and 2 mM L-glutamine at 37 °C in a CO₂ incubator.

## Animal assay

C57BL/6 mice (Jackson Laboratory, ~5 weeks old, female, strain #:000664) were used in this work. Mice were housed in Standard

Individually Ventilated Caging (IVC). The Light cycle is 12/12–12 h light/12 h dark with 7 am on–7 pm off. The Temperature is maintained between 68° and 72 °F and the humidity is between 30 and 70% per the NIH Guide. To ensure gender uniformity, the mice used in this study were all female. Also, statistical analysis is compared with or without treatment, so the sex of the host is considered less important. Therefore, the experiments were designed without considering the sex of mice. Tumour size was measured by a digital

**Fig. 6 | DTIC/Epacasome-2 fortified the PD-1 blockade to elicit tumour regression and complete metastasis remission in late-stage metastatic melanoma model. a** Drug administration timeline scheme in s.c. B16−F10-Luc2 tumour mice (*n* = 6 mice for (**a**−**g**), tumours: ~300 mm³). Mice were intravenously injected with DTIC, Epacasome-2 at eq. 75 mg DTIC/kg (based on the maximum tolerated dose (MTD) of DTIC)[51] or 41 mg EPA/kg on day 12, 14 and 16 alone or in combination with i.p. α-PD-1 (100 µg per mouse) on days 12 and 15. **b** Average tumour growth curves measured by a digital calliper. **c** Representative mice Lago bioluminescence imaging (BLI) on day 12, 15 and 18. **d** Representative ex vivo BLI for various organs on day 18. **e** Tumour metastatic rate summarised in a heatmap. **f** Normalised BLI in various organs. **g** Intratumoural IL-2 level. **h**−**m** Representative flow histogram and quantitative analysis for intratumoural Rae-1 (**h**, **i**), Mult-1 (**j**, **k**),

and MHC-I (**l**, **m**) on tumour cells (CD45⁻ cells). **n** Representative flow cytometric plots of intratumoural NKG2D⁺, CD69⁺, IFN-γ⁺, Perforin⁺ and Granzyme B⁺ associated NK⁺ cells and their respective quantification (right panel, right panel, for (**h**−**n**), *n* = 3 tumours were randomly chosen from an independent study on day 18). The flow cytometric gating strategies are placed in Supplementary Figs. 20 and 21. Intratumoural NKG2D⁺, CD69⁺, IFN-γ⁺, Perforin⁺, Granzyme B⁺, PD-1⁺, Lag-3⁺ and Tim-3⁺ associated CD8⁺ T cells flow cytometric data and gating strategies are placed in Supplementary Figs. 19 and 22. Data in (**b**, **f**, **g**, **i**, **k**, **m**, **n**, right panel) are expressed as mean ± s.d. Statistical significance was determined by one-way ANOVA followed by Tukey's multiple comparisons. Source data are provided as a Source Data file.

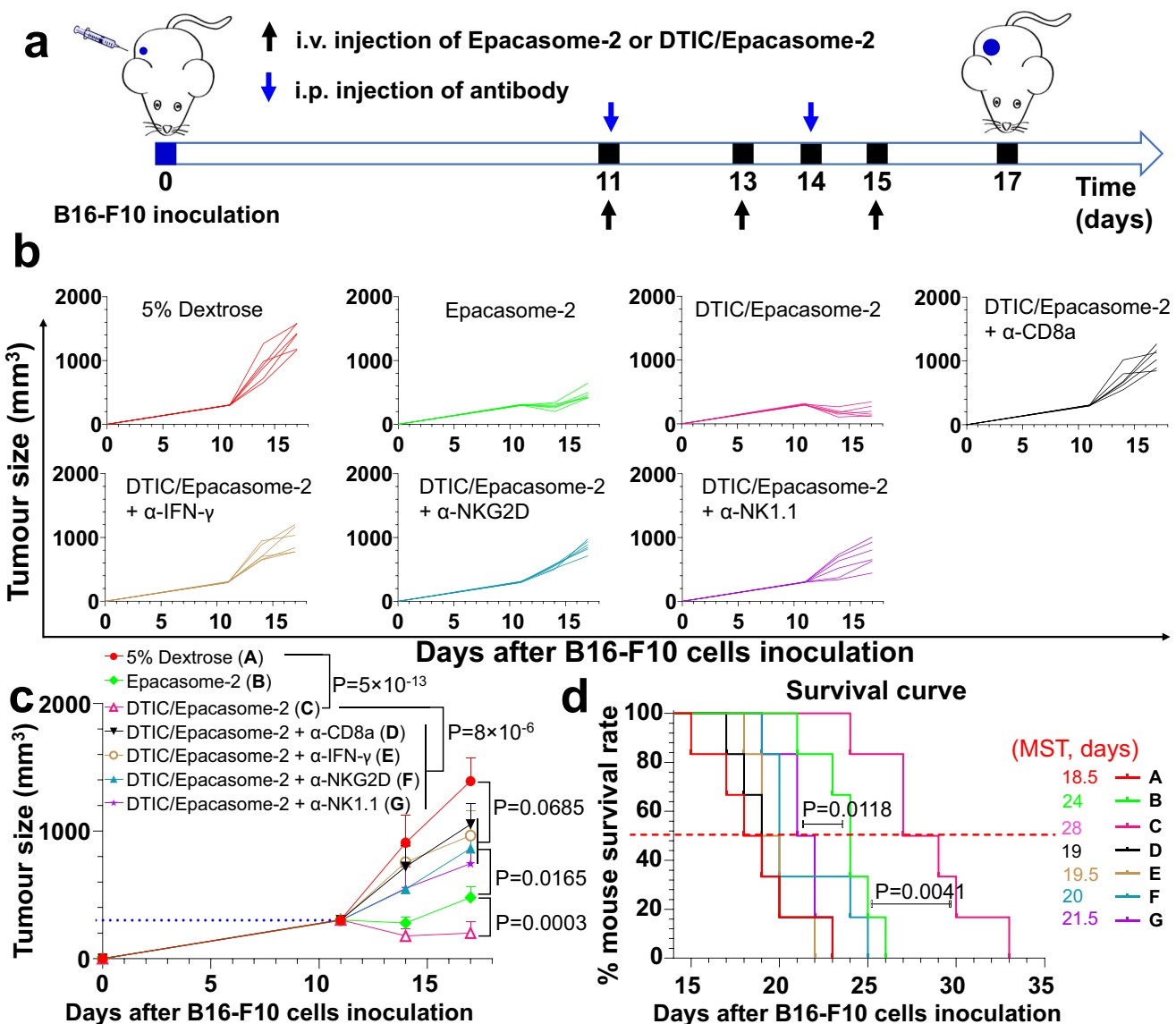

**Fig. 7 | Therapeutic effects of DTIC/Epacasome-2 are CD8⁺ T cells, IFN-γ, NK cells and NKG2D dependent. a** Drug administration timeline scheme in s.c. B16−F10 tumour mice (*n* = 6 mice, tumours: ~300 mm³). Mice were intravenously injected with DTIC, Epacasome-2 at eq. 75 mg DTIC/ or 41 mg EPA/kg on day 11, 13 and 15 alone or in combination with i.p. α-CD8a, α-IFN-γ, α-NKG2D or α-NK1.1 (200 µg per mouse) on day 11 and 14. **b**, **c** Individual (**b**) and average (**c**) tumour

growth curves measured by a digital calliper. **d** Kaplan−Meier survival curves. Data in (**c**) are expressed as mean ± s.d. (*n* = 6 mice). Statistical significance was determined by one-way ANOVA followed by Tukey's multiple comparisons test; survival curves were compared using the log-rank Mantel−Cox test. Source data are provided as a Source Data file.

calliper at indicated times and calculated according to the formula = 0.5 × length × width². Mice were removed from the respective study when tumour was found to be ≥2000 mm³ or animals became moribund with severe weight loss, extreme weakness or inactivity. The

mice were euthanized in accordance with the animal ethics guidelines of IACUC and animal welfare regulations when the tumour volume reached the maximal permitted size of 2000 mm³. However, on some occasions, the maximal permitted tumour size has been

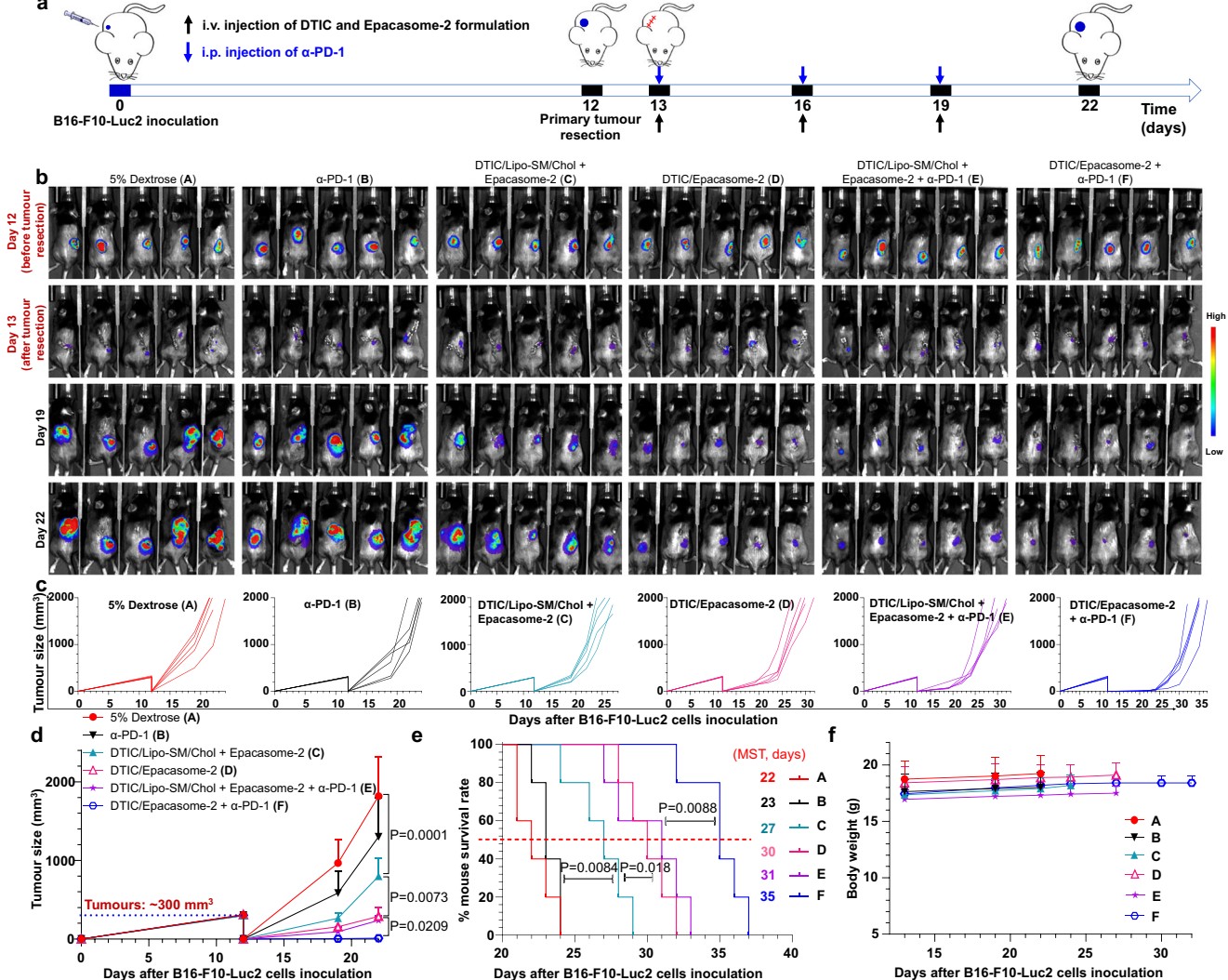

**Fig. 8 | DTIC/Epacasome-2 prevented the recurrence of melanoma tumours after surgery. a** Drug administration timeline scheme in post-surgical B16−F10-Luc2 tumour model. 12 days after the cells s.c. inoculation (*n* = 5 mice, tumours size ~300 mm³), the primary tumours were resected and mice received i.v. treatment with DTIC/Lipo-SM/Chol plus Epacasome-2 or DTIC/Ecapasome-2 at the eq. 75 mg DTIC/kg, 41 mg EPA/kg on day 13, 16 and 19 or combined with i.p. α-PD-1 (100 μg per mouse per 3 day on three occasions) on day 13, 16 and 19. **b** Mice Lago

bioluminescence imaging (BLI) on day 12 (before surgery), 13 (after surgery), 19 and 22. **c, d** Individual (**c**) and average (**d**) tumour growth curves measured by a digital calliper. **e** Kaplan−Meier survival curves. **f** Mouse body weight. Data in (**d, f**) are expressed as mean ± s.d. (*n* = 5 mice). Statistical significance in (**d**) was determined by one-way ANOVA followed by Tukey's multiple comparisons test; survival curves were compared using the log-rank Mantel−Cox test. Source data are provided as a Source Data file.

exceeded on the last day of measurement and the mice were euthanized immediately.

## The synthesis of SM-EPA
**(2 S,3 R,E)-3-((3-carboxypropanoyl)oxy)-2-palmitamidooctadec-4-en-1-yl (2-(trimethylammonio)ethyl) phosphate (SM-COOH).** The synthesis of SM-COOH was according to our previous publication[39]. 4-pyrrolidinopyridine (4-PPY, 148 mg, 1 mmol) was added to a solution of sphingomyelin (8.01 g, 10.0 mmol) and succinic anhydride (10 g, 100 mmol) in anhydrous CHCl₃ (200 mL). The solution was stirred at room temperature and monitored by TLC (developing solvent: CHCl₃/EtOH/H₂O (v/v/v, 300/200/36) with one drop of concentrated NH₃•H₂O). After completion of the reaction, CH₃OH (50 mL) was added into the mixture solution, the reaction was further stirred at room temperature for 12 h. The solvent was evaporated using rotary evaporator under vacuum, and the residue was purified by silica gel flash chromatography with CHCl₃/EtOH/H₂O (v/v/v, 300/200/36) as the elution solvent. White solid with 95% yield was achieved.

**(2 S,3 R,E)-3-((4-(((E)-N′-(3-bromo-4-fluorophenyl)-4-((2-sulfamoylamino)ethyl)amino)-1,2,5-oxadiazole-3-carboximidamido)oxy)-4-oxobutanoyl)oxy)-2-palmitamidooctadec-4-en-1-yl (2-(trimethylammonio)ethyl) phosphate (SM-EPA).** CDI (194.6 mg, 1.2 mmol) was added to a solution of SM-COOH (803.0 mg, 1.0 mmol) in anhydrous THF (40 mL). The reaction mixture was stirred at 60 °C for 48 h. After the material SM-COOH completely converted to the active intermediate via monitoring by TLC, a solution of EPA (438.2 mg, 1.0 mmol) in 10 mL of anhydrous THF was added into the reaction and further stirred at 60 °C for 48 h. The reaction was monitored by TLC, after completion of the reaction, the product was extracted by CHCl₃, the mixture was washed with 50 mM HCl aqueous solution, and then saturated brine. The organic layer was dried with anhydrous Na₂SO₄, the solvent was removed using rotary evaporator under vacuum, and the residue was purified by silica gel flash chromatography with CHCl₃/EtOH/H₂O (v/v/v, 300/200/36) as the eluting solvent. White solid with 65% yield was achieved. TLC (CHCl₃/EtOH/H₂O = 300/200/36): *R_f* = 0.32. ¹H-NMR (500 MHz, CD₃OD) δ 7.38 (dd, *J* = 6.0, 2.6 Hz, 1H), 7.15 (t, *J* = 8.6 Hz, 1H), 7.11−6.87 (m, 1H), 5.88−5.76 (m, 1H), 5.42 (dd, *J* = 15.2,

8.0 Hz, 1H), 5.38–5.32 (m, 1H), 4.32 (dd, $J = 11.6$, 4.8 Hz, 1H), 4.27 (s, 2H), 3.96–3.88 (m, 2H), 3.65–3.61 (m, 2H), 3.47 (t, $J = 5.9$ Hz, 2H), 3.29 (d, $J = 5.8$ Hz, 2H), 3.21 (d, $J = 4.5$ Hz, 9H), 2.73–2.58 (m, 4H), 2.19 (dd, $J = 14.1$, 6.9 Hz, 2H), 2.03 (dd, $J = 13.9$, 6.9 Hz, 2H), 1.61 – 1.53 (m, 2H), 1.28 (s, 46H), 0.89 (t, $J = 6.9$ Hz, 6H). $^{13}$C-NMR (126 MHz, DMSO-$d_6$) δ 172.62, 171.01, 169.72, 156.14, 154.95, 145.00, 139.80, 136.72, 136.69, 135.81, 128.79, 125.78, 116.71, 107.90, 73.93, 65.87, 58.90, 53.56, 44.01, 41.35, 35.99, 32.21, 32.15, 31.79, 29.58, 29.53, 29.51, 29.46, 29.44, 29.20, 29.17, 29.15, 28.89, 28.27, 25.80, 22.57, 14.38. HRMS (ESI) $m/z$ [M + H]$^+$ for C$_{54}$H$_{95}$BrFN$_9$O$_{12}$PS calculated 1222.57204, found 1222.57327.

### The synthesis of DTIC hydrochloride (DTIC·HCl)
DTIC (3.64 g, 20 mmol) was dissolved in 50 mL anhydrous DCM at room temperature, HCl gas was bubbled into the mixture solution for 24 h. After the completion of the reaction, the precipitation was filtered under reduced pressure, the solid was collected and dried under vacuum. White solid with 98% yield was attained. $^1$H-NMR (500 MHz, DMSO-$d_6$) δ 8.96 (s, 1H), 8.02 (s, 1H), 7.13 (s, 1H), 3.65 (s, 3H), 3.32 (s, 3H). $^{13}$C-NMR (126 MHz, DMSO-$d_6$) δ 159.47, 141.97, 133.83, 114.29, 44.71, 37.82. HRMS (ESI) $m/z$ [M + H]$^+$ for C$_6$H$_{11}$N$_6$O calculated 183.09889, found 183.09884.

### Drug release kinetics of DTIC/Epacasome-2
PBS (pH = 7.4, pH = 6.5, pH = 5.5 or esterase enzyme (10 units, # 46069, Sigma-Aldrich)) containing 0.5% (w/v) Tween 80 were used as the release medium for the drug release kinetics study of Epacasome-2 in vitro. Briefly, two mL of Epacasome-2 (2 mg EPA/mL) were transferred into dialysis tubes (MWCO = 12 kDa, Spectrum Laboratories), and then suspended in 200 mL release medium inside a beaker shielded with aluminium foil. The temperature of the release medium was kept at 37 °C and stirred at 100 rpm. At indicated time points, 5 and 100 μL of solution were taken from the dialysis tubes and dialysate, respectively. The concentration of SM-EPA and EPA remaining in the dialysis tubes and dialysate were measured by HPLC with the detector set at 254 nm.

### Antitumour efficacy analysis
The median survival time (MST) was the time point at which the mice survival rate equals 50%, time to reach endpoint (TTE) was the time point for tumour size reaching 2000 mm$^3$ or moribund criteria[62]. Tumour growth delay (TGD) was defined as the time differences between treated mouse and mean of control mice that tumour size reached to 1000 mm$^3$, the time point of nontreated tumour growing to 1000 mm$^3$ was determined by fitting each growth curve of the control tumours with an exponential function, the individual time of each treated tumours to reach 1000 mm$^3$ was calculated from linear interpolation of the two nearest points below and above the 1000 mm$^3$ (see ref. [63]). The increase in live span [ILS, Eq. (1)] was calculated as a percentage using the following equation[64]:

$$\frac{\text{survival time of treated mice} - \text{median survival time of control mice}}{\text{median survival time of control mice}} \times 100\% \tag{1}$$

### Immunohistochemistry (IHC) and antibodies
The procedure of IHC staining was according to our previously established protocols[39]. Briefly, the dissected tumour blocks were fixed in 4% paraformaldehyde for 12 h, and then embedded in paraffin. In all, the 4-μm thickness of sections was processed for the tumour blocks, the slices were loaded onto positively charged glass slides by the UArizona Cancer Center TACMASR Core facility for a series of IHC staining processes and procedures following our previous established protocols. Anti-interferon gamma (ab9657, 1/200), anti-CD8α (ab209775, 1/100), anti-granzyme B (ab4059, 1/100) and anti-perforin (ab16074, 1/600) were purchased from Abcam; anti-NK1.1 (# MA1-

70100, 1/100) and anti-CD69 (# PA5-102562, 1/100) were obtained from Invitrogen; anti-NKG2D (# BS-0938R, 1/100) and anti-Foxp3 (#12653 S, 1/100) were purchased from Bioss and Cell Signaling, respectively. After staining, the slide sections were dried and observed under Olympus VS200 slide scanner, the data were analysed by Olympus image viewer (version: OlyVIA V4.1).

### Preparation of Epacasome and EPA physical laden liposomes
According to the ratio listed in Fig. 2a, SM, Chol, DSPE-PEG$_{2K}$ and SM-EPA conjugate (for Cy5/Epacasome, 0.2% w/w of DSPE-Cy5 was added), were dissolved in ethanol by a 100 mL round bottom glass flask. The organic solvent was evaporated under reduced pressure by using a rotatory evaporator (RV 10 digital, IKA®) to generate a thin film, and then further dried under ultra-high vacuum (MaximaDry, Fisherbrand) for 0.5 h. The lipid film was hydrated by 5% dextrose at 60 °C for 30 min, and then sonicated under an ice bath for 12 min by using a pulse 3/2 s on/off at a power output of 60 W (VCX130, Sonics & Materials Inc). The nanovesicles were further purified by an ultra-centrifugation at 100,000× g for 45 min to remove any unencapsulated SM-EPA. EPA physically laden liposomes and empty liposome were prepared according to the ratio listed in Supplementary Fig. 5c and reported methods[44,45], the molar ratio of SM in EPA/Lipo-SM/Chol = "SM + SM-EPA" in Epacasome-2. The size, zeta potential, PDI, morphology and EPA content in the nanovesicles were determined by DLS, Cryo-EM and HPLC respectively. EPA drug loading capacity [DLC, Eq. (2)] was calculated as below:

$$\frac{\text{weight of EPA in nanovesicles}}{\text{weight of total lipids}} \times 100\% \tag{2}$$

### Loading DTIC into Epacasome-2 to prepare DITC/Epacasome-2
SM, SM-EPA conjugate, Chol and DSPE-PEG$_{2K}$ at the molar ratio of 62.63/20.19/12.78/4.40 were dissolved in ethanol by a 100 mL round bottom glass flask. DTIC/Lipo-SM/Chol consisted of 82.82% SM, 12.78% Chol, 4.40% DSPE-PEG$_{2K}$ (molar ratio). The organic solvent was evaporated under reduced pressure by using a rotatory evaporator (RV 10 digital, IKA®) to generate a thin film, and then further dried under ultra-high vacuum (MaximaDry, Fisherbrand) for 0.5 h. The lipid film was hydrated by 95 or 250 mg/mL DTIC solution (in DI water) at 60 °C for 30 min, and then sonicated under an ice bath for 12 min by using a pulse 3/2 s on/off at a power output of 60 W (VCX130, Sonics & Materials Inc). The nanovesicles were further purified by running through a PD-10 column to remove the unencapsulated DTIC with PBS as the eluent. The size, zeta potential, PDI, morphology and DTIC content in the nanovesicles were determined by DLS, Cryo-EM and HPLC respectively. The DTIC drug loading capacity [DLC, Eq. (3)] and drug loading efficiency [DLE, Eq. (4)] were calculated as below:

$$\frac{\text{weight of encapsulated drug}}{\text{weight of (total lipids + encapsulated drug)}} \times 100\% \tag{3}$$

$$\frac{\text{weight of encapsulated drug}}{\text{weight of input drug}} \times 100\% \tag{4}$$

### IDO inhibitory effect in HeLa and 4T1 cells
The inhibition of IDO in HeLa and 4T1 cells was measured by an in vitro IDO assay[29]. Briefly, HeLa or 4T1 cells were seeded in 96-well plates at a density of $5 \times 10^3$ cells/well and incubated overnight. Recombinant human or mouse IFN-γ (final concentration 50 ng/mL) and various concentrations (0, 1, 5, 10, 20, 50, 100, 200 and 10000 nM equivalent EPA) of free EPA, Lipo-SM/Chol, EPA/Lipo-SM/Chol, or Epacasome-2 were added into the wells. After the cells were incubated at 37 °C for 48 h, 150 μL of the supernatants from each well was transferred to a

new 96-well plate. To hydrolyse N-formylkynurenine to kynurenine, 75 μL of 30% trichloroacetic acid was added into each well and the mixture solution was incubated at 50 °C for 30 min. The solution was mixed with equal volume of Ehrlich reagent (2% p-dimethylamino-benzaldehyde w/v in glacial acetic acid) and incubated at ambient temperature for 10 min. The intensity of the solution was detected at 490 nm by a SpectraMax M3 reader (SoftMax Pro (v. 7.1.0), Molecular Devices).

## T-cell proliferation by co-culture

B16−F10 cells were stimulated by IFN-γ (50 ng/mL) for 48 h to induce IDO1 expression, then treated by 10 μg/mL Mitomycin C for 30 min before co-culture[46,65]. Splenocytes from C57BL/6 mice were produced by passing through nylon wool columns after lysing red blood cells. IFN-γ-stimulated B16−F10 cells ($1 \times 10^5$ cells/well) were mixed with splenocytes ($5 \times 10^5$ cells/well, pre-stained with CFSE). Free EPA, Lipo-SM/Chol, EPA/Lipo-SM/Chol or Epacasome-2 (0, 8, 40, 200 and 1000 nM equivalent EPA) were then added to cells at eq. dose of EPA. To assess T-cell proliferation, anti-CD3 (100 ng/mL) and mouse recombinant IL-2 (10 ng/mL) were added to co-cultures. After 72 or 96 h, CD8$^+$ T-cell proliferation was determined by FACS analysis.

## Cellular uptake of Epacasome-2

B16−F10, RAW 264.7 or 4T1 cells were seeded in a 175-cm² flask at the density $5 \times 10^7$ cells/flask and incubated overnight. Free EPA EPA/Lipo-SM/Chol, or Epacasome-2 with eq. 10 μM were added into the flask, respectively. After incubating at 37 °C for 24 or 48 h, the cells were trypsinized, washed with cold PBS two times, calculated the cells number, homogenised and digested in methanol for 2 h. After centrifuge, the concentration of EPA and SM-EPA in the supernatant were measured by HPLC.

## Cellular uptake of Cy5/Epacasome-2 and the potential internalisation pathways

B16−F10 cells were seeded in 12-well plates at the density $1 \times 10^5$/well and incubated overnight. The cells were pre-incubated with six endocytosis inhibitors for 30 min, chlorpromazine (50 μM, clathrin-mediated endocytosis inhibitor), genistein (200 μM, caveolae-mediated endocytosis inhibitor), methyl-β-cyclodextrin (Me-β-CD, 800 μM, caveolae-mediated endocytosis inhibitor), wortmannin (5 μM, macropinocytosis inhibitor), cytochalasin D (5 μM, macropinocytosis inhibitor), and NaN$_3$ (10 mM, energy-dependent endocytosis inhibitor). After that, the cells were incubated with Cy5/Epacasome-2 (50 μg EPA/mL) in the presence of the endocytosis inhibitors throughout the 2-h uptake experiment at 37 °C, respectively. Another independent assay was incubated the cells with Cy5/Epacasome-2 at 4 °C or 37 °C for 2 h, respectively. Finally, the cells were trypsinized, washed with cold PBS two times, resuspended in 400 μL of staining buffer and analysed by a BD FACSCanto™ II flow cytometer (BD Bioscience).

## Intracellular trafficking of Cy5/Epacasome-2 in B16−F10 cells

B16−F10 cells were seeded on the coverslip in 6-well plates at a density of $1 \times 10^6$ cells/well and incubated overnight. The cells were incubated with Cy5/Epacasome-2 (50 μg EPA/mL) for 2 h. The cells were counterstained with Hoechst 33342 (Invitrogen, #H3570) for cell nucleus and LysoView™ 488 (Biotium, #70067-T) for lysosome following the manufacturer's instructions. The coverslips were mounted on glass microscope slides with a drop of anti-fade mounting media (Sigma-Aldrich, USA). The cellular localisation was visualised under a laser scanning confocal microscope (Leica LAS-AF software (v. 2.7.3.9723)) at the University of Arizona Cancer Center (UACC) Tissue Acquisition and Cellular/Molecular Analysis Shared Resource (TACMASR).

## Combination index (CI) study

The drug combination study of SM-EPA with DTIC in B16−F10 cells were performed according to the previous method[66]. Briefly, 12 h after the cells seeded in 96-well plates at the density of 2000 cells/well, the combination molar ratio of 1:1, 2:1, 3:1, 3.66:1, 4:1, 5:1, 6:1, 1:2, 1:3, 1:4, 1:5, 1:6 and the drugs alone (drug concentration: 200, 100, 50, 25, 12.5, 6.25, 3.12, 1.56, 0.78 and 0.39 μM) were added into the wells and incubated for 48 h, respectively. The IC$_{50}$ values of each compound were calculated by the cell viabilities that were measured by MTT. The combination index [CI, Eq. (5)] plot depicts the IC$_{50}$ values of each compound at different combination ratios by using the following equation:

$$\frac{\text{IC}_{50}\ \text{of SM} - \text{EPA}}{\text{IC}_{50}\ \text{of SM} - \text{EPA alone}} + \frac{\text{IC}_{50}\ \text{of DTIC}}{\text{IC}_{50}\ \text{of DTIC alone}} \qquad (5)$$

CI < 1 indicates the synergy, CI = 1 indicates the additive effect, CI > 1 indicates the antagonism.

## Pharmacokinetics and biodistribution

Free EPA (10 mg/kg, in 10% dimethylacetamidein saline), EPA/Lipo-SM/Chol (10 mg EPA/kg), Epacasome-2 (equivalent of 10 mg EPA/kg), free DTIC (75 mg/kg, DTIC·HCl dissolved in 5% dextrose), DTIC/Lipo-SM/Chol (75 mg/kg), Epacasome-2 (41 mg EPA/kg) plus DTIC/Lipo-SM/Chol (75 mg/kg) or DTIC/Epacasome-2 (75 mg DTIC/kg, 41 mg EPA/kg) were intravenously injected into B16−F10 tumour-bearing mice ($n = 3$ mice, tumour size: ~400 mm³) via tail vein. At 0.083, 0.333, 1, 2.5, 8, 12 and 24 h after drug administration, blood was withdrawn; and plasma was collected by using a plasma tube (BD Microtainer) and then digested in methanol prior to HPLC measurement for the EPA and SM-EPA, respectively. At 24 h after drug administration, tumour tissues and major organs (heart, liver, spleen, lung, kidneys) were collected and homogenised in acidified methanol (0.075 M HCl, 900 μL/100 mg tissue) before HPLC drug content analysis. The pharmacokinetic parameters of free EPA, DTIC, DTIC/Lipo-SM/Chol, Epacasome-2 and DTIC/Epacasome-2 were assessed by using PKSolver software (version 2.0)[67].

## Visualisation of Cy5/Epacasome-2 deep tumour penetration

To visualise the deep extravasation and penetration of the nanovesicles inside the tumour, Cy5/Epacasome-2 (equivalent of 10 mg EPA/kg, 0.2% w/w of DSPE-Cy5) or Cy5/EPA/Lipo-SM/Chol (equivalent of 10 mg EPA/kg, 0.2% w/w of DSPE-Cy5) were intravenously injected into B16−F10 tumour-bearing mice ($n = 3$ mice, tumour weight: ~300 mg) via tail vein. 24 h after the injection, tumours were dissected and frozen in an acetone−dry ice mixture followed by immunofluorescence examination. The blood vessels in the tumours were marked with a primary anti-CD31 (also known as PECAM1) antibody (ab28364, 1:50, Abcam) followed by an Alexa Fluor 488-conjugated secondary antibody (ab150073, 1:400, Abcam). DAPI (4,6-diamidino-2-phenylindole) was used to stain the cellular nuclei[39]. The fluorescence signals were visualised under a Zeiss LSM880 inverted confocal microscope (Zen Black software (v. 14.022.021)) at the Innovation and Impact's Imaging Core-Optical Core Facility at the University of Arizona.

## Conversion the human dose of EPA to mouse equivalent dose

Conversion the human dose of EPA to mouse equivalent dose was according to the guidance of Center for Drug Evaluation and Research (CDER) from US FDA[68]. The conversion information is listed in Supplementary Table 4.

## Therapeutic efficacy study of Epacasome-2 in B16−F10 melanoma tumour model

In total, $1 \times 10^5$ of B16−F10 cells in 100 μL of serum-free DMED medium were subcutaneously injected into C57BL/6 mice ($n = 6$ mice). For the

efficacy study in Fig. 4, when tumours reached ~100 mm³, mice received p.o. (in 0.5% methylcellulose (MC) aqueous solution) administration of free EPA (41 mg/kg) or i.v. (in 10% dimethylacetamidein saline) administration of 5% dextrose (set as vehicle control), free EPA (41 mg/kg) or Epacasome-2 (equivalent of 41 mg EPA/kg) on day 8, 10, 12 and 14 alone or combined with i.p. α-PD-1 (100 µg per mouse per 3 day on three occasions) from the day 8. Primary tumours were measured by digital calliper and tumour-bearing mice were photographed on day 15. The mice body weight and survival were closely monitored as indicated. In an independent efficacy study ($n = 6$ mice), the primary tumours were dissected on day 15 for measuring the concentration of Trp and Kyn by ELISA. The blood was collected in lithium heparin tubes (BD Microtainer™) and the plasma were subjected to measure the concentration of Trp and Kyn by ELISA. In another independent efficacy study ($n = 6$ mice), the primary tumours were dissected on the same day and three tumours were randomly chosen for the flow cytometry study.

In an independent study, healthy C57BL/6 mice ($n = 6$ mice) received the same treatment as above, on day 15, the blood was withdrawn in lithium heparin tubes (BD Microtainer™) and the serum was submitted to University Animal Care Pathology Services Core at UArizona for serum chemistry analysis (Liasys 330). The whole blood was collected in dipotassium EDTA tube (BD Microtainer™) and used for leukocytes, erythrocytes, and thrombocytes analysis (Hemavet 950FS). The liver, spleen, kidneys, muscle, and intestines were collected and soaked in 4% paraformaldehyde overnight prior to H&E staining and periodic acid-Schiff (PAS) reaction and counterstained with hematoxylin, respectively.

In the efficacy study of Supplementary Figs. 14 and 15, when tumours reached ~100 mm³, mice ($n = 6$ mice) received i.v. administration of 5% dextrose (set as vehicle control), Lipo-SM/Chol (empty liposome, same equivalent as EPA/Lipo-SM/Chol without loading EPA), EPA/Lipo-SM/Chol (41 mg EPA/kg), EPA/Lipo-DOPE/Chol (41 mg EPA/kg), EPA/Lipo-HSPC/Chol (41 mg EPA/kg), or Epacasome-2 (equivalent of 41 mg EPA/kg) on day 8, 10, 12 and 14 alone or combined with i.p. α-PD-1 (100 µg per mouse per 3 day on three occasions) from the day 8. The mice body weight and survival were closely monitored as indicated. On day 15, the primary tumours were dissected and divided into two pieces, one part for IHC study, another part for measuring the concentration of Trp and Kyn by ELISA. The blood was collected in lithium heparin tubes (BD Microtainer™) and the plasma were subjected to measure the concentration of Trp and Kyn by ELISA.

## Measurements of Trp and Kyn concentration in tumour tissues and plasma
The dissected tumour from each mouse was homogenised in 4 °C PBS (900 µL/100 mg tissue), respectively. After centrifuged at 2000 × g under 4 °C for 5 min, the supernatant was collected and used to measure the Trp and Kyn concentration by a kynurenine/tryptophan ratio ELISA pack (ImmuSmol, ISE-2227) following the manufacturer's protocol. For the plasma, it was directly measured by the ELISA pack according to the manufacturer's protocol.

## Therapeutic efficacy study of DTIC/Epacasome-2 in B16–F10-Luc2 and B16–F10 melanoma tumour model
In all, $1 × 10^5$ of B16–F10-Luc2 cells in 100 µL of serum-free DMED medium were subcutaneously injected into C57BL/6 mice ($n = 6$ mice). When tumours reached ~300 mm³, mice received IV administration of 5% dextrose (set as vehicle control), free DTIC (75 mg/kg, DTIC·HCl dissolved in 5% dextrose), Epacasome-2 (equivalent of 41 mg EPA/kg), DTIC/Lipo-SM/Chol (75 mg DTIC/kg), Epacasome-2 (41 mg EPA/kg) combined with DTIC/Lipo-SM/Chol (75 mg DTIC/kg), DTIC/Epacasome-2 (75 mg DTIC/kg, 41 mg EPA/kg) on days 12, 14 and 16 alone or combined with i.p. α-PD-1 (BioXCell, clone RMP1-14, 100 µg per mouse per 3 day on two occasions) on day 12 and 15[39,69]. Tumour

burden on mouse whole-body was imaged by Lago optical imaging after intraperitoneally injected with 150 mg/kg D-Luciferin (GoldBio, MO, USA) on day 12, 15 and 18. Primary tumours were measured by digital calliper. On day 18, following injection of D-Luciferin, mice were dissected after 5 min, the tumour metastasis in the major organs was measured and analysed by ex vivo Lago imaging. The primary tumours were collected and dissected into two pieces, one part for ELISA, and another part for IHC study. To investigate the intratumoural immune response elicited by the treatment, a separate efficacy study was performed. The tumours were dissected on day 18 after receiving the same treatments as above, three tumours were randomly chosen and subjected to flow cytometry study. In another independent study, B16–F10-Luc2 tumour-bearing mice received the same treatment as above, tumour growth, mice body weight and survival were closely monitored as indicated.

In an independent study, healthy C57BL/6 mice ($n = 6$ mice) received the same treatment as above, on day 18, the blood was withdrawn in lithium heparin tubes (BD Microtainer™) and serum submitted to University Animal Care Pathology Services Core at UArizona for serum chemistry analysis (Liasys 330). The whole blood was collected in dipotassium EDTA tube (BD Microtainer™) and used for leukocytes, erythrocytes, and thrombocytes analysis (Hemavet 950FS).

For efficacy study in Fig. 7, $1 × 10^5$ of B16–F10 cells in 100 µL of serum-free DMED medium were subcutaneously injected into C57BL/6 mice ($n = 6$ mice). When tumours reached ~300 mm³, mice received IV administration of 5% dextrose (set as vehicle control), Epacasome-2 (equivalent of 41 mg EPA/kg), DTIC/Epacasome-2 (75 mg DTIC/kg, 41 mg EPA/kg) on day day 11, 13 and 15 alone or combined with i.p. α-CD8a (BioXCell, clone 53-6.7), α-IFN-γ (BioXcell, clone R4-6A2), α-NKG2D (BioXCell, clone HMG2D) and α-NK1.1 (BioXcell, clone PK136) on day 11 and 14 (200 µg per mouse per 3 day on two occasions). Primary tumours were measured by digital calliper and tumour-bearing mice were photographed on day 17, mice body weight and survival were closely monitored as indicated. To confirm the depletion of immune cell subsets by the antibodies, an independent efficacy study was performed following the same treatment as above. The tumours were dissected on day 17 and subject to IHC study.

In the sequential combination treatments of Supplementary Fig. 25, when tumours reached ~200 mm³ ($n = 6$ mice), mice received i.v. administration of 5% dextrose (vehicle control), Epacasome-2 (equivalent of 41 mg EPA/kg), DTIC (75 mg DTIC/kg) and i.p. α-PD-1 (BioXCell, clone RMP1-14, 100 µg per mouse) at indicated time point, tumour growth, mice body weight and survival were closely monitored.

## Measurements of IL-2 levels in tumour tissues
The dissected tumour from each mouse was homogenised in 4 °C PBS (900 µL/100 mg tissue), respectively. After centrifuging at 2000 × g under 4 °C for 5 min, the supernatant was collected and used to measure the IL-2 concentration by IL-2 mouse ELISA kit (Invitrogen, #BMS601) following the manufacturer's protocol.

## Therapeutic efficacy study of DTIC/Epacasome-2 in surgical B16–F10-Luc2 melanoma tumour model
To evaluate the effects of DTIC/Epacasome-2 on melanoma recurrence, $1 × 10^5$ of B16–F10-Luc2 cells in 100 µL of serum-free DMED medium were subcutaneously injected into C57BL/6 mice ($n = 5$ mice). Twelve days after the inoculation (tumours size ~300 mm³), the primary tumours were then resected by sterile instruments under anaesthesia, with ~1% residual tumour tissue leaving behind to mimic the presence of residual microtumours in the surgical bed[54]. The wound was closed using an Autoclip wound closing system (BD Autoclip™, 22-275998). Twenty-four hours after the surgery, mice received IV administration of 5% dextrose (set as vehicle control),

Epacasome-2 (41 mg EPA/kg) combined with DTIC/Lipo-SM/Chol (75 mg DTIC/kg), DTIC/Epacasome-2 (equivalent of 75 mg DTIC/kg, 41 mg EPA/kg) on day 13, 16 and 19 alone or combined with i.p. α-PD-1 (100 μg per mouse per 3 day on three occasions) on day 13, 16 and 19. Tumour burden on mouse whole-body was imaged by Lago optical imaging after intraperitoneally injected with 150 mg/kg D-Luciferin (GoldBio, MO, USA) on day 12, 13, 19 and 22. The primary tumours were measured by digital calliper, mice body weight and survival were closely monitored as indicated.

## Flow cytometry analysis
The dissected tumours were cut into small pieces by scissors on ice, and then digested in DMEM medium (0.5 mg/mL collagenase type I, Worthington Biochemical Corporation) for 1 h at 37 °C. To obtain single cell for analysis, the samples were meshed by a 70-μM cell strainer twice. The cell solution was incubated with Ack lysing buffer (Gibco, 2217610) to lyse the red blood cells according to the manufacturer's protocols. The cell samples were washed with 4 °C PBS twice and resuspended in 4 °C staining buffer. After counting and aliquoting the cell, the cell suspensions were pre-incubated with FcBlock (TruStain fcXTM anti-mouse CD16/32, clone 93, 101320, BioLegend, 0.5 μg/100 μL) at 4 °C for 20 min to avoid nonspecific binding. Cells were stained with fluorescence-labelled antibodies at 4 °C for 30 min and live cells were gated using Zombie. LAG-3(CD223)-APC (C9B7W, #562346, 1/100), CD45-APC-Cy™7 (30-F11, #557659, 1/100), CD8a-PE (53-6.7, #561095, 1/100), CD11c-PerCP-Cy5.5 (HL3, #560584, 1/100), CD80-APC (16-10A1, #553766, 1/100), CD86-PE (GL1, #553692,1/100), CD3-APC-eFluor 780 (17A2, #47-0032-82, 1/100) and CD25-APC (PC61.5, #17-0251-82, 1/100) were purchased from BD Biosceinces. Foxp3-PE (MF23, #563101, 1/100), Mult-1 (5D10, # 5013229, 1/100, goat anti-Armenian Hamster IgG (H + L) secondary antibody-PE #5010771, 1 μg/sample) and Granzyme B-eFluor 660 (NGZB, #50-8898-82, 1/100) were purchased from eBiosciences. Rae-1-PE (186107, #FAB17582P, 1/100) was purchased from Bio-Techne. Zombie Violet™ Fixable Viability Kit (#423114), Tim-3-Alexa Fluor® 647 (RMT3-23, #119744, 1/100), PD-1 (CD279)-APC (29 F.1A12, #135210, 1/100), CD4-Alexa Fluor 488 (RM4-5, #100529, 1/100), perforin-APC (S16009B, #154304, 1/100), NKG2D-APC (CX5, #130212, 1/100), CD69-APC (H1.2F3, #104514, 1/100), Gr-1-APC (RB6-8C5, #108412, 1/100), CD206-APC (C068C2, #141708, 1/100), F4/80-FITC (BM8, #123108, 1/100), CD11b-PE (M1/70, #101208, 1/100), NK1.1-PE (S17016D, #156504, 1/100), MHC class I-Alexa Fluor® 647 (AF6-88.5, #116512, 1/100) and IFN-γ-APC (XMG1.2, #505810, 1/100) were purchased from BioLegend. Multi-parameter staining was used to measure the T cells: (i) NKG2D$^+$ T cells (CD3$^+$/CD8$^+$/NKG2D$^+$), (ii) CD69$^+$ T cells (CD3$^+$/CD8$^+$/CD69$^+$), (iii) IFN-γ$^+$ T cells (CD3$^+$/CD8$^+$/IFN-γ$^+$), (iv) granzyme B$^+$ T cells (CD3$^+$/CD8$^+$/granzyme B$^+$), (v) perforin$^+$ T cells (CD3$^+$/CD8$^+$/perforin$^+$), (vi) Tregs (CD4$^+$/CD3$^+$/CD25$^+$/Foxp3$^+$), exhausted T cells (i) PD-1$^+$ T cells (CD3$^+$/CD8$^+$/PD-1$^+$), (ii) Lag-3$^+$ T cells (CD3$^+$/CD8$^+$/Lag-3$^+$), (iii) Tim-3$^+$ T cells (CD3$^+$/CD8$^+$/Tim-3$^+$), NK cells: (i) NKG2D$^+$ NK cells (CD3$^-$/NK1.1$^+$/NKG2D$^+$), (ii) CD69$^+$ NK cells (CD3$^-$/NK1.1$^+$/CD69$^+$), (iii) IFN-γ$^+$ NK cells (CD3$^-$/NK1.1$^+$/IFN-γ$^+$), (iv) granzyme B$^+$ NK cells (CD3$^-$/NK1.1$^+$/granzyme B$^+$), (v) perforin$^+$ NK cells (CD3$^-$/NK1.1$^+$/perforin$^+$), DCs cells: (i) CD80$^+$/CD86$^+$ DCs (CD45$^+$/CD11c$^+$/CD80$^+$/CD86$^+$), (ii) CD11c$^+$/CD103$^+$ DCs (CD45$^+$/CD11c$^+$/CD103$^+$), TAMs cells: M1-type (CD45$^+$/CD11b$^+$/F4/80$^+$/CD206$^-$) and M2-type (CD45$^+$/CD11b$^+$/F4/80$^+$/CD206$^+$), MDSC cells: G-MDSC (CD45$^+$/CD11b$^+$/Gr-1$^{high}$) and M-MDSC (CD45$^+$/CD11b$^+$/Gr-1$^{int}$), tumour cells: (i) MHC-I tumour cells (CD45$^-$/MHC-I$^+$), (ii) Rae-1 tumour cells (CD45$^-$/Rae-1$^+$), (iii) Mult-1 tumour cells (CD45$^-$/Mult-1$^+$). For IFN-γ staining, cells were first stimulated with cell stimulation cocktail (eBioscience, #00-4975-93) for 4 h prior to extracellular and intracellular staining according to the manufacturer's protocol. Cells were fixed and permeabilized using a staining Buffer Set (eBioscience, #00-5523-00) followed by

intracellular staining of IFN-γ, granzyme B, perforin or Foxp3. After washing, cells were measured on flow cytometry and analysed by FlowJo software (version 10.0.7, TreeStar). The numbers presented in the flow cytometry analysis images were percentage-based.

## Statistical analysis
The level of significance in all statistical analyses was set at $P < 0.05$. Data are presented as mean ± s.d. and were analysed using the two-tailed, unpaired Student's $t$ test for two groups or one-way analysis of variance (ANOVA) for three or more groups followed by Tukey's multiple comparisons test using Prism 8.0 (GraphPad Software). Kaplan–Meier survival curves were compared with the log-rank Mantel–Cox test.

## Reporting summary
Further information on research design is available in the Nature Portfolio Reporting Summary linked to this article.

## Data availability
All the data supporting the findings of this study are available within the Article, Supplementary Information or Source Data file. Source data are provided with this paper.

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

## Acknowledgements

This work was supported in part by a Startup Fund from the R. Ken Coit College of Pharmacy at The University of Arizona (UArizona) and a PhRMA Foundation for Research Starter Grant in Drug Delivery, and by National Institutes of Health (NIH) grants (R35GM147002 and R01CA272487) to J.L. (R01 CA229418) to G.W and (K08 CA276137-01A1) to J.E. We acknowledge the use of Mass Spectrometry in Analytical and Biological Mass Spectrometry Core Facility at the UArizona BIO5 Institute; the UArizona Translational Bioimaging Resource Core for the Lago live animal imaging; Tissue Acquisition and Cellular/Molecular Analysis Shared Resource (TACMASR) at UArizona Cancer Center (UACC) for the immunofluorescence staining, we thank Patty Jansma, manager of the Office of Research, Innovation and Impact's Imaging Core-Optical Core Facility at the University of AZ for providing training and support; UArizona University Animal Care Pathology Services for the serum chemistry and haematological counts; and the Flow Cytometry Immune Monitoring Shared Resource (FCIMSR) core for flow cytometry studies at UACC, which are supported by NIEHS P30 ES006694 and NCI P30 CA023074. We also thank the Arizona State University's John Cowley Center for Hight Resolution Electron Microscopy (the specific instrumentation used was supported by the NSF, MRI grant NSF1531991) for Cryo-EM.

## Author contributions

J.L. conceived and supervised the project. J.L. and Z.W. designed the experiments, analysed the data, and wrote the manuscript. Z.W. performed the experiments. W.L. assisted in IDO1 inhibition and T-cell proliferation assay. Y.J., T.B.T., L.E.C., J.C and M.K. assisted in nanoparticle preparation, sample preparation for HPLC analysis, and in vivo animal studies. G.W. helped with data interpretation and offered comments on the manuscript. J.E. contributed to the clinical research insights on the post-surgical melanoma model and provided remarks/revisions for the manuscript from the clinical perspective. All the authors discussed the results and commented on the manuscript.

## Competing interests

The authors declare the following. J.L. has applied for patents related to this technology (patent title: Immunogenic nanovesicles for cancer immunotherapy, patent number: WO2022115488). The remaining authors declare no competing interests.
