## [Peer Review File · Nature Communications]

Sphingomyelin-derived nanovesicles for the delivery of the IDO1 inhibitor epacadostat enhance metastatic and post surgical melanoma immunotherapyREVIEWER COMMENTS

Reviewer #1 (Remarks to the Author): with expertise in nanotechnology, cancer therapy

Summary

This manuscript developed a sphingomyelin (SM)-derived epacadostat (EPA) nanovesicles to rescue this IDO inhibitor from the failure of Phase III clinical trial of EPA combining with ICB therapies by improving the pharmacokinetics and tumor accumulation. A chemically conjugating strategy attaching EPA to SM via a hydrolase-responsive oxime-ester bond was employed to load EPA to the lipid-based vesicle, termed Epacosome, with a high encapsulation efficiency. Benefiting from the favorable delivery performance, the authors proved that the Epacosome showed a more robust potentiating effect with PD-1 blockade than free EPA, leading to more potent anticancer efficacy in the B16-F10 melanoma model. Additionally, after co-encapsulation with immunogenic dacarbazine, Epacosome was demonstrated to boost the PD-1 inhibitor with complete metastasis remission in late-stage melanoma mice via upregulating the NKG2D-mediated cytotoxic T lymphocytes and natural killer cells responses.

General comments

The authors constructed a drug delivery system to solve the translational dilemma of EPA in combination with ICB therapies, which has been previously published in Nat. Nano. From the perspectives of both Epacosome and DTIC/Epacosome-2, the authors provided sufficient results to prove that this strategy could pave a new way for IDO inhibitors to enter the clinic. But there are some major questions, especially about the translational potential and treatment regimens, that should be addressed.

Major comments

1. It is not sure if this proposed formulation, Epacosome, could reliably improve the translational potential of EPA.

1) Why did this study not employ lipid-based nanoparticles to physically encapsulate EPA if the goal is just to improve its delivery performance? Although the authors adopted a sophisticated conjugation strategy to load EPA into a lipid-based nanoparticle, the newly synthesized SM-EPA actually complicated the translation process of EPA based on

considering the clinical application of liposome and the supportive results of EPA itself in Phase I and II clinical trials. Besides, in the parts of loading DTIC to the nanoparticle, the results demonstrated the physical encapsulation could also realize high loading efficiency and favorable delivery behavior.

2) The authors conjugated EPA to SM through a hydrolase-sensitive oxime-ester bond. But there are not enough in vitro and in vivo data to establish the advantages of this fabrication for EPA delivery. Specially, it would be better if a similar lipid-based nanoparticle with physical encapsulation of EPA was selected as a control to evaluate the delivery performance.

2. The efficacy of immunotherapy-based combination anticancer treatments in the clinic is usually closely related to the administration regimen. The reason why the author chose the co-administration of DTIC and EPA (even and α -PD-1) was not well clarified. Given that these three drugs act on the different stages of the immune response, have the authors considered sequential combination treatments or other more optimized treatment regimens?

Minor comments:

1. The results in Fig. 3e are not enough to demonstrate the enhanced blood vessel extravasation and deep penetration of Epacosome in tumor tissue. Additional proofs and even more control groups should be supplemented to confirm these advantages caused by Epacosome.

2. As for Fig. 5d and 5e, since the ratios of the combined drugs always play an important role in the final treatment efficacy, did the author consider screening the optimal ratio of DTIC and EPA in DTIC/Epacosome-2 based on their synergistic effect?

3. In the treatment efficacy evaluation parts, an administration scheme might improve the readability of the relevant figures.

4. About Fig. 2g, the incubation time should be indicated in the caption or the relevant interpretation.

Reviewer #2 (Remarks to the Author): with expertise in nanotechnology, cancer therapy

The authors reported an EPA nanovesicle therapeutic platform (Epacasome) by chemically attaching EPA to sphingomyelin via a hydrolase-highly responsive oxime-ester bond.

Epacasome improved pharmacokinetics and fortifies tumor accumulation with efficient intratumoral drug release and deep tumor penetration. The authors concluded that Epacasome increased significantly antitumor effects and immune responses with minimized metastasis. Although, they have done all necessary experiments to support their idea and the manuscript is very well-organized, there are some major concerns regarding the study which should be answered before being considered for publication.

Major comments

1. The most important concern is the authors didn't include an empty liposome as a control group in their study. Control liposome is a very important group in their study because they have used Sphingomyelin (SM) in the formulation. SM, a major membrane sphingolipid, is the precursor of bioactive products. It was proved that sphingolipid metabolism is a dynamic process and sphingolipid metabolites such as ceramide, sphingosine, and sphingosine 1-phosphate (S1P) are now recognized as messengers playing essential roles in cell growth and survival. It is clear now that signal transduction through SM and ceramide strongly affects the immune response generation either directly through cell signaling events or indirectly through cytokines produced by other cells as the result of signaling through the SM pathway (1-4). Therefore, it is difficult to predict whether these results are related to the immunomodulatory properties of SM or not.

2. There are at least two published papers (5-6) that have used EPA in liposomal form with the same purpose in cancer therapy. The authors did not mention them in their manuscript and they didn't compare them with Epacasome. What is the advantage of this study compared to theirs?

3. The authors have synthesized a new molecule i.e., EPA-SM, bridged by an oxime ester bond which is sensitive to the hydrolase in tumors. The authors claimed that the hydrolase cleavages EPA-SM to EPA and SM, but SM could be also hydrolyzed to sphingosine, fatty acid, and phosphorylcholine. How do they confirm the result would be only SM and EPA, not other possible options? The other concern is Hydrolases are found in the liver, erythrocytes,

muscle, kidney, intestinal mucosa, etc., and hydrolyze aliphatic and aromatic esters. This means that EPA-SM can be cleaved in other sites and we still have an off-target effect with EPA. How can the authors address this issue?

Minor comments

Method

1. In the section IDO inhibitory effect in Hela cells, the authors just mentioned “the various concentrations of free EPA or Epacosome-2”. Please mention the concentrations that were used.
2. In the T cell proliferation assay similar to IDO inhibitory assay, the exact dose of EPA and Epacosome-2 did not mention in the method. Why did not the authors check the proliferation of T cells over a longer period of time? Or in the different time points?
3. In evaluating the therapeutic efficacy of Epacosome-2 in B16-F10 melanoma tumor model, it is better to have a timeline to exactly show the exact time of treatment. This makes it easier to follow for the reader. Also, the same for the healthy mice. The B16-F10 cells injected in DMEM? Or PBS?

Results

1. In figure 4, the way that authors used for labeling different groups is confusing. The significancy on the tumor growth curve (fig 4c) and survival (fig 4d) results are not clear, especially the tumor growth curve it's hard to follow.
2. It is better to present the results of MST as a separate table and analyze the significancy between groups. It will also be more informative if the authors present the results of TTE (time to reach endpoint), TGD (tumor growth delay), and ILS (increased live span) for each treatment group and compare the results to show the significant differences.
3. Since melanoma is an aggressive type of cancer and induction of an effective immune response is hard to achieve, it will be also valuable to check the level of exhausted CD8 T cells (i.e., PD-1, Tim-3, and Lag-3).
4. In presenting the results of figure 6 in the body of the article, the author presented graphs a-e and then jumped to g-l and again f, m. Please keep them in order.
5. In Figure 7, the DTIC/Epacosome-2 showed the smallest tumor size among all treatment groups, but the significance in fig 7b is not clear. Does it mean that all the groups were compared to DTIC/Epacosome-2 group?

6. Again, the significance in figure 8c, d is not clear. The table of TTE, TGD, MST, and ILS results also works for the combination therapy with PD-1 mAb study.

7. Please clarify which day was considered as the end of the study in combination therapy study. In the individual growth curve in each group (fig 8b) day 22 was shown, in tumor size (fig 8c) day 32 and in survival curve (fig 8d) day 37? Since all mice in both dextrose and ant PD-1 groups were died by day 22, the author can use border line on day 22 in fig 8b for all treatment groups to show the number and the tumor growth of mice in treatment groups till end of study that seems to be day 37.

8. It would be noteworthy if the authors could do IHC or mIF for visualizing the infiltrated T cells and reduced Tregs, NK cells, ... other immune cells in the tumor site of different treatment groups.

Discussion

It is more Like the results part and used figures. The authors should discuss more and compare with previous studies.

References:

1. Melendez AJ. Sphingosine kinase signaling in immune cells: Potential as novel therapeutic targets. *Biochim Biophys Acta*. 2008;1784:66–75.
2. Kolesnick R. The therapeutic potential of modulating the ceramide/sphingomyelin pathway. *J Clin Invest* . 2002;110:3–8.
3. Ballou LR, Lauderkind SJ, Rosloniec EF, Raghov R. Ceramide signalling and the immune response. *Biochim Biophys Acta*. 1996;1301:273–287.
4. Nixon, Graeme F. Sphingolipids in inflammation: pathological implications and potential therapeutic targets. *British journal of pharmacology*. 2009;158:982-993.
5. Tahaghoghi-Hajghorbani S, Khoshkhabar R, Rafiei A, et al. Development of a novel formulation method to prepare liposomal Epcadostat. *Eur J Pharm Sci*. 2021;165:105954.
6. Chen Y, Du Q, Zou Y, et al. Co-delivery of doxorubicin and epacadostat via heparin coated pH-sensitive liposomes to suppress the lung metastasis of melanoma. *Int J Pharm*. 2020;584:119446.

Reviewer #3 (Remarks to the Author): with expertise in IDO targeting, cancer immunology/immunotherapy

The manuscript by Wang and Li et al. entitled “Sphingomyelin-derived epacadostat nanovesicles for enhanced metastatic and post-surgical melanoma immunotherapy” describes the use of sphingomyelin nanovesicles to enhance uptake of epacadostat, an IDO1 inhibitor, to improve anti-tumor response. They highlight the reduced in vivo bioavailability and tumor uptake of epacadostat as one of the reasons for poor clinical outcome and a rationale for designing of a nanovesicle therapeutic delivery. Assessing blood kinetics and intratumoral uptake using HPLC, they showed dramatic intratumoral delivery of epacasome over free epacadostat delivery with similar tissue biodistribution. Immunofluorescence of tumor sections with labeled epacasome confirmed marked tumor localization. Authors also combine epacadostat with DTIC front line drug treatment to enhance combinatorial anti-tumor affect. In subsequent tumor implantation and immunophenotype experiments, authors claim survival and antitumor benefits of there epacasome over epacadostat and other therapeutic modalities including anti-PD-1 immune checkpoint. While this study shows nice pharmacodynamics and claims of antitumor effect from epacasome, there are some major lapses in experimental design and major concerns in conclusions made that must be addressed.

Main points/questions:

1. Fig 2: B16 cells don't express high IDO1. Please use alternative tumors such as 4T1 or B16 that overexpress IDO1 to validate claim of epacasome mediated IDO1 inhibition.
2. Fig 2: Authors show different epacasome iterations but do not have a negative control (with empty nanovesicle or non-active compound) in addition to epacasome. This is needed to differentiate effect of nanovesicle affect over the delivery of epacadostat.
3. Fig 3: Liver expresses high levels of IDO1 and TDO2. There is an increase in liver and spleen epacasome uptake more so than other organs. Tissue damage including looking at necropsy, is needed to assess potential toxicity.
4. Extended Data 3 and 5: Authors need to reanalyze flow cytometry data to include flow plots that gate on live cells and subsequent distinct population gating in addition to FMO.

The concern is nonviable cells and debris is being gated on that skew analysis. It would probably important to seek the expertise of someone who has deeper expertise in flow cytometry to help with this aspect.

5. Figure 6B: Subplot in tumor measurements does not have X and Y labels and not clear as to what the plot is referring to.

6. Fig 7: Confirmation of depletion of immune cell subsets needs to be addressed.

Minor points

1. Fig 2: Use macrophage for uptake in addition to tumor.

2. Fig 4: panel B and G is unnecessary remove to supplement keep the summary bar plots.

3. Reference to Figures is out of order of appearance.

4. Fig 6: Missing survival data

5. Justification for anti-PD1 at 100ug dose typically 200ug – 250ug and reference clone used.

6. More rigor is needed in figure preparation and paper write up. Multiple grammar and misspelling make reading manuscript difficult, careful editing needed. Misspelling of epacosome, MDSC, etc. This is unfortunate as the premise of this work is great.

7. Multiple figures of tumor bearing image of mice (Fig. 4B, 6D, 8A) need to be shown in qualitative measurements with representative images placed in supplement.

Reviewer #1:

Major comments

Q1. It is not sure if this proposed formulation, Epacosome, could reliably improve the translational potential of EPA.

Q1) Why did this study not employ lipid-based nanoparticles to physically encapsulate EPA if the goal is just to improve its delivery performance? Although the authors adopted a sophisticated conjugation strategy to load EPA into a lipid-based nanoparticle, the new-synthesized SM-EPA actually complicated the translation process of EPA based on considering the clinical application of liposome and the supportive results of EPA itself in Phase I and II clinical trials. Besides, in the parts of loading DTIC to the nanoparticle, the results demonstrated the physical encapsulation could also realize high loading efficiency and favorable delivery behavior.

Response: Thank you for the critical comment. We agree with the reviewer on the necessity of including the lipid-based nanoparticles to physically encapsulate EPA as control. As suggested, we have used liposome consisting of SM/Chol/DSPE-PEG_{2K} (Lipo-SM/Chol, at the same lipid ratio as Epacosome-2, SM in EPA/Lipo-SM/Chol = “SM + SM-EPA” in Epacosome-2) to physically load EPA. In addition, as suggested by reviewer-2, physically loading EPA into the liposomes composed of DOPE/Chol/DSPE-PEG_{2K} (Lipo-DOPE/Chol) and HSPC/Chol/DSPE-PEG_{2K} (Lipo-HSPC/Chol) have also been compared according to the published literature^{1,2}. Our results showed that the EPA loading capacity was around 0.23-1.27% through physically encapsulating EPA into various liposomes, which were significantly increased in Epacosome (8.19-16.38%); and physically loaded EPA/Lipo also had poorer stability than Epacosome (**Fig. 2a** and **Supplementary Fig. 5**). More importantly, Epacosome-2 markedly improved the pharmacokinetics and tumour uptake, as well as enhanced the antitumour efficacy compared to the physically loaded EPA/Lipo (**Fig. 3** and **Supplementary Figs. 14,15**). To clarify, direct encapsulation of DTIC into Epacosome-2 only met with limited success (DLC: 0.24%). To

facilitate DTIC loading, it was converted into a salt form, DTIC·HCl (Fig. 5d). This strategy resulted in much higher DLC (up to 40.13%) in DTIC/Epacosome-2 (Fig. 5g). We also attempted to use EPA·HCl for direct loading into liposome but failed because EPA·HCl was not stable as it readily underwent a Beckmann rearrangement and generated inactive urea derivative in acidic conditions. Taken together, these findings justified the use of Epacosome to improve the translational potential of EPA.

a

Epacosome	SM (molar %)	SM-EPA (molar %)	Cholesterol (molar %)	DSPE-PEG _{2K} (molar %)	EPA loading capacity (weight %)	DLS Size by intensity (d.nm)	Zeta Potential (mV)	Polydispersity Index (PDI)
Epacosome-1	67.61	15.70	12.42	4.28	8.19	129.7 ± 5.74	-30.8 ± 6.48	0.238 ± 0.010
Epacosome-2	62.63	20.19	12.78	4.40	10.24	106.2 ± 9.50	-33.8 ± 4.71	0.082 ± 0.019
Epacosome-3	57.34	24.96	13.16	4.53	12.28	119.2 ± 4.98	-32.1 ± 7.21	0.249 ± 0.021
Epacosome-4	51.73	30.02	13.57	4.68	14.33	115.7 ± 4.58	-28.7 ± 8.02	0.236 ± 0.016
Epacosome-5	45.76	35.41	14.00	4.83	16.38	116.2 ± 6.81	-30.5 ± 5.34	0.243 ± 0.011

Fig. 2a. A table illustrating the physicochemical properties of Epacosome composed of SM/SM-EPA/Cholesterol/DSPE-PEG_{2K} with various molar ratios.

Supplementary Fig. 5. The DLS Size (a) and zeta potential (b) of Epacosome monitoring over a 15-day period at 4 °C. c, A table illustrating the physicochemical properties of Lipo-SM/Chol (empty liposome), EPA/Lipo-SM/Chol, EPA/Lipo-DOPE/Chol and EPA/Lipo-HSPC/Chol with detailed ratios^{1,2}. d.n.m, diameter values in nanometres. DLS, dynamic light scattering. d, DLS size distribution by intensity. The DLS Size (e) and zeta potential (f) of various liposomes monitoring over a 15-day period at 4 °C. Data in a, b, c (right portion), e and f are represented as mean ± s.d. (n = 3).

Fig. 5d,e,g. d, Synthesis of DTIC·HCl to facilitate the loading of DTIC to Epacosome-2. e, Schematic to show the manufacturing process, g, a table delineating the physicochemical characterizations.

2) The authors conjugated EPA to SM through a hydrolase-sensitive oxime-ester bond. But there are not enough in vitro and in vivo data to establish the advantages of this fabrication for EPA delivery. Specially, it would be better if a similar lipid-based nanoparticle with physical encapsulation of EPA was selected as a control to evaluate the delivery performance.

Response: We would like to thank the reviewer for the valuable comment. We agree with reviewer on setting similar lipid-based nanoparticle with physical encapsulation of EPA as a control. We have included EPA physically laden liposome, EPA/Lipo-SM/Chol composed of SM/Chol/DSPE-PEG_{2K} at the same lipid ratio as Epacosome-2 (SM in EPA/Lipo-SM/Chol = “SM + SM-EPA” in Epacosome-2) as a control (**Supplementary Fig. 5c**). We showed that Epacosome-2 significantly increased cellular uptake, and enhanced IDO1 inhibition and CD8⁺ T cells proliferation in vitro compared to EPA/Lipo-SM/Chol (**Fig. 2d-f**). The pharmacokinetics and biodistribution studies demonstrated that Epacosome-2 improved the half-life of EPA from ~0.73 h in EPA/Lipo-SM/Chol to ~4.72 h and delivered 2.6-fold more EPA to tumours (**Fig. 3a-c**). The confocal imaging with DSPE-Cy5 labeling further corroborated that Epacosome-2 enabled better tumour penetration vs EPA/Lipo-SM/Chol (**Fig. 3e**). In addition, Epacosome-2 significantly outperformed EPA/Lipo-SM/Chol in suppressing B16-F10 tumour growth, inhibiting IDO1 activity, and eliciting the antitumour immune responses in vivo (**Supplementary Fig. 14**). Furthermore, as suggested by reviewer-2, we also included EPA/Lipo-DOPE/Chol and EPA/Lipo-HSPC/Chol^{1,2}, two literature reported liposomes with physically encapsulated EPA, as additional controls. Similarly, Epacosome-2 beat EPA/Lipo-DOPE/Chol and EPA/Lipo-HSPC/Chol on anticancer

efficacy and immune responses (Supplementary Fig. 15). These results substantiated the advantages of using Epacosome-2 to deliver EPA over physical encapsulation of EPA in liposomes.

Fig. 2d-f. **d**, IDO1 enzymatic inhibitory activity via measuring the Kyn in supernatants in HeLa cells treated with IFN- γ along with free EPA, Lipo-SM/Chol, EPA/Lipo-SM/Chol and Epacosome-2 at equivalent (eq.) EPA concentration (0, 1, 5, 10, 20, 50, 100, 200 and 10000 nM.) for 48 h. **e**, T cell proliferation by co-culture^{3,4}. B16-F10 cells were stimulated by IFN- γ to induce IDO1 expression, then treated by Mitomycin C prior to mixing with splenocytes. Free EPA, EPA/Lipo-SM/Chol and Epacosome-2 were added to co-culture cells at eq. dose of EPA. To evaluate T cell proliferation, anti-CD3 and IL-2 were added to co-cultures. 3 days later, CD8⁺ T cell proliferation was assessed by FACS analysis. **f**, Cellular uptake levels and EPA release of Epacosome-2 in B16-F10 cells after 24 and 48 h incubation measured by HPLC, respectively.

Fig. 3. a-d, Blood kinetics (**a**), biodistribution (**b**), and EPA intratumoural release (**c**) at 24 h in mice from (**a**) and various pharmacokinetic parameters in B16-F10 tumour-bearing C57BL/6 mice ($n=3$; tumour: $\sim 400 \text{ mm}^3$) following a single i.v. administration of free EPA, EPA/Lipo-SM/Chol and Epacosome-2 at eq. 10 mg EPA/kg. The per cent injected doses in Epacosome represents the released EPA and SM-conjugated EPA. Drug contents in plasma and major tissues were measured by HPLC (**Supplementary Fig. 3**). **e**, Unraveling the potential of Epacosome-2 for extravasating and penetrating the tumours post an i.v. injection into mice bearing subcutaneous B16-F10 tumours ($n=3$ mice, tumours, $\sim 400 \text{ mm}^3$). 24 h after injecting Epacosome (red), CLSM of sections of B16-F10 tumours was conducted. Cell nuclei were stained by DAPI (blue). Blood vessels were stained with PECAM1 antibody followed by Alexa Fluor 488 secondary

antibody staining (green). Scale bars, 50 μm . Data in **a-d** are expressed as mean \pm s.d (n = 3). Statistical significance was determined by one-way ANOVA followed by Tukey's multiple comparisons test.

Supplementary Fig. 14. **a**, Drug administration timeline scheme of antitumour efficacy in subcutaneous (s.c.) B16-F10 tumour model (n=6, tumours: \sim 100 mm³), mice i.v. injected with Lipo-SM/Chol (empty liposome), EPA/Lipo-SM/Chol, Epacosome-2 at eq. 41 mg EPA/kg on day 8, 10, 12 and 14 alone or combined with i.p. α -PD-1 (BioXCel, clone RMP1-14, 100 μg per mouse per 3 day for 3 times) from day 8. **b**, Individual tumour growth curves. **c-d**, Average tumour size growth curves (**c**) and body weight (**d**). **e,f**, The ratio of Trp (nM)/kyn (nM) concentration in plasma (**e**) and tumours (**f**). **g**, Mice bearing s.c. B16-F10 tumour images taken on day 15. **h**) Representative IHC staining for CD8, Granzyme B, IFN- γ , and Foxp3 on day 15, scale bar = 100 μm . Data in **c-f** are expressed as mean \pm s.d. Statistical significance was determined by one-way ANOVA followed by Tukey's multiple comparisons test.

Supplementary Fig. 15. **a**, Drug administration timeline scheme of antitumour efficacy in subcutaneous (s.c.) B16-F10 tumour model ($n=6$, tumours: ~ 100 mm³), mice i.v. injected with EPA/Lipo-DOPE/Chol, EPA/Lipo-HSPC/Chol, Epacosome-2 at eq. 41 mg EPA/kg on day 8, 10, 12 and 14 from day 8. **b**, Individual tumour growth curves. **c-d**, Average tumour size growth curves (**c**) and body weight (**d**). **e, f**, The ratio of Trp (nM)/kyn (nM) concentration in plasma (**e**) and tumours (**f**). **g**, Mice bearing s.c. B16-F10 tumour images taken on day 15. **h**) Representative IHC staining for

CD8, Granzyme B, IFN- γ , and Foxp3 on day 15, scale bar = 100 μ m. Data in **c-f** are expressed as mean \pm s.d. Statistical significance was determined by one-way ANOVA followed by Tukey's multiple comparisons test.

Q2. The efficacy of immunotherapy-based combination anticancer treatments in the clinic is usually closely related to the administration regimen. The reason why the author chose the co-administration of DTIC and EPA (even and α -PD-1) was not well clarified. Given that these three drugs act on the different stages of the immune response, have the authors considered sequential combination treatments or other more optimized treatment regimens?

Response: Thanks for your constructive comments and providing us with the insights on how to further improve our manuscript. As suggested by reviewer, we have performed comprehensive sequential combination treatments by adjusting drug administration sequence (**Supplementary Fig. 21a**) in comparison with co-administration regimen. Our results elucidated that co-administration regimen (group H) delivered the best anticancer efficacy and longest survival time.

Supplementary Fig. 21. **a**, Drug administration table scheme of antitumour efficacy in subcutaneous (s.c.) B16-F10 tumour model ($n=6$, tumours: ~ 200 mm³), mice injected with Epacosome-2 (i.v. at eq. 41 mg EPA/kg), DTIC (i.v. 75 mg/kg) and combined with i.p. α -PD-1 (BioXCell, clone RMP1-14, 100 μ g per mouse) from day 11. **b-c**, Average tumour size growth curves (**b**) and body weight (**c**). **d**, Kaplan-Meier survival curves. **e**, Mice bearing s.c. B16-F10 tumour images taken on day 15. Data in **b-c** are expressed as mean \pm s.d. Statistical significance was determined by one-way ANOVA followed by Tukey's multiple comparisons test; survival curves were compared using the log-rank Mantel-Cox test.

Minor comments:

Q1. The results in Fig. 3e are not enough to demonstrate the enhanced blood vessel extravasation and deep penetration of Epacosome in tumour tissue. Additional proofs and even more control groups should be supplemented to confirm these advantages caused by Epacosome.

Response: Thank you for your comment. As requested, we have included DSPE-Cy5 labelled EPA/Lipo-SM/Chol as an additional control in this study. Our results (Fig. 3e) unraveled that Cy5/Epacosome-2 extravasated farther away from blood vessels with deeper tumour penetration as compared to Cy5/EPA/Lipo-SM/Chol. This further confirmed the enhanced blood vessel extravasation and deep penetration of Epacosome-2 in tumour tissue.

Fig. 3e. Unraveling the potential of Epacosome-2 for extravasating and penetrating the tumours post an i.v. injection into mice bearing subcutaneous B16-F10 tumours ($n = 3$ mice, tumours, $\sim 400 \text{ mm}^3$). 24 h after injecting Epacosome (red), CLSM of sections of B16-F10 tumours was conducted. Cell nuclei were stained by DAPI (blue). Blood vessels were stained with PECAM1 antibody followed by Alexa Fluor 488 secondary antibody staining (green). Scale bars, 50 μm .

Q2. As for Fig. 5d and 5e, since the ratios of the combined drugs always play an important role in the final treatment efficacy, did the author consider screening the optimal ratio of DTIC and EPA in DTIC/Epacosome-2 based on their synergistic effect?

Response: Thank you for the critical comment. As suggested, we have systemically screened and evaluated the optimal ratio of DTIC and SM-EPA based on the synergistic effect in B16-F10 cells (**Fig. 5a-c**). The combination index (CI) revealed that DTIC/SM-EPA with molar ratio of 3/1 and 4/1 yielded the lowest CI values (0.40 and 0.41, respectively), which suggested the potent synergy in inhibiting B16-F10 cells proliferation. Considering the maximum tolerated dose for DTIC, the fixed amount of EPA on the bilayer of Epacosome-2 as well as the EPA dose for animal efficacy studies converted from the clinical application^{5,6}, the molar ratio of 3.66/1 (DTIC/SM-EPA, 75 mg DTIC/kg and 41 mg EPA/kg) were considered to be the most appropriate and were used for the efficacy studies. We also demonstrated that the CI of 3.66/1 (DTIC/SM-EPA) is 0.40, confirming this ratio is optimal.

Fig. 5a-c, investigation of the optimal drug ratios in B16-F10 by MTT assay. **a,b**, Cells were treated at various drug combination ratios for 48 h. **c**, the combination index (CI) was calculated based on the IC₅₀ values of individual drug.

Q3. In the treatment efficacy evaluation parts, an administration scheme might improve the readability of the relevant figures.

Response: We would like to thank the reviewer for the valuable comment. As suggested by reviewer, we have included a detailed administration scheme in **Figs. 4a,6a,7a,8a, Extended Data Figs. 2a,4a, and Supplementary Figs. 14a,15a,21a**.

Q4. About Fig. 2g, the incubation time should be indicated in the caption or the relevant interpretation.

Response: Thank you for your comment. The incubation time was 2 h in **Fig. 2g**, which has been indicated in the caption and Method section of the revised manuscript.

Reviewer #2:

Major comments

Q1. The most important concern is the authors didn't include an empty liposome as a control group in their study. Control liposome is a very important group in their study because they have used Sphingomyelin (SM) in the formulation. SM, a major membrane sphingolipid, is the precursor of bioactive products. It was proved that sphingolipid metabolism is a dynamic process and sphingolipid metabolites such as ceramide, sphingosine, and sphingosine 1-phosphate (S1P) are now recognized as messengers playing essential roles in cell growth and survival. It is clear now that signal transduction through SM and ceramide strongly affects the immune response generation either directly through cell signaling events or indirectly through cytokines produced by other cells as the result of signaling through the SM pathway (1-4). Therefore, it is difficult to predict whether these results are related to the immunomodulatory properties of SM or not.

Response: We would like to thank the reviewer for the valuable comment. We agree with reviewer on the importance of setting empty liposome as a control to evaluate the impact from SM. We have used SM to prepare the empty liposome, Lipo-SM/Chol consisting of SM/Cholesterol (Chol)/DSPE-PEG_{2K} with the same lipid ratio as Epacosome-2 (SM in Lipo-SM/Chol = "SM + SM-EPA" in Epacosome-2) (**Supplementary Fig. 5c**). We demonstrated that Lipo-SM/Chol had no effect on IDO1 inhibition and did not render any benefits for CD8⁺ T cells proliferation (**Fig. 2d,e**). Furthermore, compared to vehicle control, Lipo-SM/Chol exerted negligible inhibition on tumour growth in B16-F10 tumour model and had no significant effects on the Trp/Kyn ratio in both plasma and tumours as well as the antitumour immune responses (**Supplementary Fig. 14**). In sum, these findings demonstrated that the observed antitumour effects of Epacosome-2 was not related to the immunomodulatory properties of SM.

C

Liposomes	Lipid ratio	EPA loading efficiency (%)	EPA loading capacity (weight %)	DLS Size by intensity (d.nm)	Zeta Potential (mV)	Polydispersity Index (PDI)
Lipo-SM/Chol (empty liposome)	SM:Chol:DSPE-PEG _{2K} = 82.82/12.78/4.40 (molar ratio)	NA	NA	120.2 ± 3.06	-10.3 ± 2.65	0.110 ± 0.024
EPA/Lipo-SM/Chol	SM:Chol:DSPE-PEG _{2K} = 82.82/12.78/4.40 (molar ratio)	72.3	1.27	154.5 ± 7.53	-18.5 ± 4.83	0.114 ± 0.016
EPA/Lipo-DOPE/Chol	DOPE:DOTAP:Chol:DSPE-PEG _{2K} = 9/9/2/1 (weight ratio)	62.1	1.08	126.6 ± 4.98	-16.6 ± 6.82	0.228 ± 0.018
EPA/Lipo-HSPC/Chol	HSPC:Chol:DSPE-PEG _{2K} = 22.3/7.3/7.3 (weight ratio)	53.6	0.23	117.7 ± 5.03	-12.6 ± 3.67	0.115 ± 0.010

Supplementary Fig. 5c. A table illustrating the physicochemical properties of Lipo-SM/Chol (empty liposome), EPA/Lipo-SM/Chol, EPA/Lipo-DOPE/Chol and EPA/Lipo-HSPC/Chol with detailed ratios^{1,2}. d.nm, diameter values in nanometres. DLS, dynamic light scattering.

Fig. 2d,e. **d**, IDO1 enzymatic inhibitory activity via measuring the Kyn in supernatants in HeLa cells treated with IFN- γ along with free EPA, EPA/Lipo-SM/Chol and Epacasome-2 at equivalent (eq.) EPA concentration (0, 1, 5, 10, 20, 50, 100, 200 and 10000 nM) for 48 h. **e**, T cell proliferation by co-culture^{3,4}. B16-F10 cells were stimulated by IFN- γ to induce IDO1 expression, then treated by Mitomycin C prior to mixing with splenocytes. Free EPA, EPA/Lipo-SM/Chol and Epacasome-2 were added to co-culture cells at eq. dose of EPA. To evaluate T cell proliferation, anti-CD3 and IL-2 were added to co-cultures. 3 days later, CD8⁺ T cell proliferation was assessed by FACS analysis.

Q2. There are at least two published papers (5-6) that have used EPA in liposomal form with the same purpose in cancer therapy. The authors did not mention them in their manuscript and they didn't compare them with Epacasome. What is the advantage of this study compared to theirs?

Response: Thank you for the constructive comment. Sorry for the oversight and as advised by reviewer, we have cited these two papers in the revised manuscript. We also agree with the reviewer on the necessity of including these two physically encapsulated EPA liposomes (EPA/Lipo-DOPE/Chol and EPA/Lipo-HSPC/Chol) as controls^{1,2}. To that end, we have prepared these two EPA/Lipo with the same lipid ratios reported in literature (**Supplementary Fig. 5c**)^{1,2}. Our results showed that while both EPA/Lipo-DOPE/Chol and EPA/Lipo-HSPC/Chol exhibited marked antitumour efficacy and immune responses compared to vehicle control, which were far more superior in our Epacasome-2 (**Supplementary Fig. 15**). Apart from these two EPA/Lipo, as suggested by reviewer-1, we also compared Epacasome-2 with EPA/Lipo-SM/Chol (EPA physically encapsulated in liposome composed of SM/Chol/DSPE-PEG_{2K} with the same lipid ratio as Epacasome-2). We found that Epacasome-2 outperformed EPA/Lipo-SM/Chol in suppressing tumour growth and eliciting anticancer immunity with improved pharmacokinetics and tumour uptake/penetration (**Fig. 3** and **Supplementary Fig. 14**).

Supplementary Fig. 15. **a**, Drug administration timeline scheme of antitumour efficacy in subcutaneous (s.c.) B16-F10 tumour model ($n=6$, tumours: $\sim 100 \text{ mm}^3$), mice i.v. injected with EPA/Lipo-DOPE/Chol, EPA/Lipo-HSPC/Chol, Epacosome-2 at eq. 41 mg EPA/kg on day 8, 10, 12 and 14 from day 8. **b**, Individual tumour growth curves. **c-d**, Average tumour size growth curves (**c**) and body weight (**d**). **e,f**, The ratio of Trp (nM)/kyn (nM) concentration in plasma (**e**) and tumours (**f**). **g**, Mice bearing s.c. B16-F10 tumour images taken on day 15. **h** Representative IHC staining for CD8, Granzyme B, IFN- γ , and Foxp3 on day 15, scale bar = 100 μm . Data in **c-f** are expressed as mean \pm s.d. Statistical significance was determined by one-way ANOVA followed by Tukey's multiple comparisons test.

Fig. 3. Improved pharmacokinetics and tumour delivery with efficient EPA release and deep penetration in tumours. **a-d**, Blood kinetics (**a**), biodistribution (**b**), and EPA intratumoural release (**c**) at 24 h in mice from (**a**) and various pharmacokinetic parameters in B16-F10 tumour-bearing C57BL/6 mice ($n=3$; tumour: $\sim 400 \text{ mm}^3$) following a single i.v. administration of free EPA, EPA/Lipo-SM/Chol and Epacosome-2 at eq. 10 mg EPA/kg. The per cent injected doses in Epacosome represents the released EPA and SM-conjugated EPA. Drug contents in plasma and major tissues were measured by HPLC (**Supplementary Fig. 3**). **e**, Unraveling the potential of Epacosome-2 for extravasating and penetrating the tumours post an i.v. injection into mice bearing subcutaneous B16-F10 tumours ($n = 3$ mice, tumours, $\sim 400 \text{ mm}^3$). 24 h after injecting Epacosome (red), CLSM of sections of B16-F10 tumours was conducted. Cell nuclei were stained by DAPI (blue). Blood vessels were stained with PECAM1 antibody followed by Alexa Fluor 488 secondary

antibody staining (green). Scale bars, 50 μm . Data in **a-d** are expressed as mean \pm s.d. ($n = 3$). Statistical significance was determined by one-way ANOVA followed by Tukey's multiple comparisons test.

Supplementary Fig. 14. **a**, Drug administration timeline scheme of antitumour efficacy in subcutaneous (s.c.) B16-F10 tumour model ($n=6$, tumours: $\sim 100 \text{ mm}^3$), mice i.v. injected with Lipo-SM/Chol (empty liposome), EPA/Lipo-SM/Chol, Epacosome-2 at eq. 41 mg EPA/kg on day 8, 10, 12 and 14 alone or combined with i.p. α -PD-1 (BioXCell, clone RMP1-14, 100 μg per mouse per 3 day for 3 times) from day 8. **b**, Individual tumour growth curves. **c-d**, Average tumour size growth curves (**c**) and body weight (**d**). **e,f**, The ratio of Trp (nM)/kyn (nM) concentration in plasma (**e**) and tumours (**f**). **g**, Mice bearing s.c. B16-F10 tumour images taken on day 15. **(h)** Representative IHC staining for CD8, Granzyme B, IFN- γ , and Foxp3 on day 15, scale bar = 100 μm . Data in **c-f** are expressed as mean \pm s.d. Statistical significance was determined by one-way ANOVA followed by Tukey's multiple comparisons test.

Q3. The authors have synthesized a new molecule i.e., EPA-SM, bridged by an oxime ester bond which is sensitive to the hydrolase in tumours. The authors claimed that the hydrolase cleavages EPA-SM to EPA and SM, but SM could be also hydrolyzed to sphingosine, fatty acid, and phosphorylcholine. How do they confirm the result would be only SM and EPA, not other possible options? The other concern is Hydrolases are found in the liver, erythrocytes, muscle, kidney, intestinal mucosa, etc., and hydrolyze aliphatic and aromatic esters. This means that EPA-SM can be cleaved in other sites and we still have an off-target effect with EPA. How can the authors address this issue?

Response: Thank you for the constructive comment. To clarify, the focus is the EPA release from SM-EPA as which will exert the proposed antitumour responses. We are aware of the potential metabolites from the SM. To rule out the possible immunostimulatory properties from SM metabolism, we used SM to prepare the empty liposome, Lipo-SM/Chol with the same lipid ratio as Epacosome-2 (**Supplementary Fig. 5c**). We demonstrated that Lipo-SM/Chol had no effect on IDO1 inhibition and did not render any benefits for CD8⁺ T cells proliferation (**Fig. 2d,e**). Furthermore, compared to vehicle control, Lipo-SM/Chol exerted negligible inhibition on the tumour growth in B16-F10 tumour model and had no significant effects on the Trp/Kyn ratio in both plasma and tumours as well as the antitumour immune responses (**Supplementary Fig. 14**). In sum, these findings demonstrated that the observed antitumour effects of Epacosome-2 was not related to the immunomodulatory properties of SM or its metabolites.

To investigate the possible off-target effect of Epacosome-2, healthy mice received the same treatments as in **Fig. 4a**. We confirmed that compared to vehicle control, there were no overt toxicities in erythrocytes, leukocytes, thrombocytes, liver, spleen, kidney, muscle as well as intestinal mucosa in Epacosome-2 treated group (**Extended Data Fig. 2**). These results corroborated the excellent in vivo safety profiles of Epacosome-2.

Fig. 2d,e. **d**, IDO1 enzymatic inhibitory activity via measuring the Kyn in supernatants in HeLa cells treated with IFN- γ along with free EPA, EPA/Lipo-SM/Chol and Epacosome-2 at equivalent (eq.) EPA concentration (0, 1, 5, 10, 20, 50, 100, 200 and 10000 nM) for 48 h. **e**, T cell proliferation by co-culture^{3,4}. B16-F10 cells were stimulated by IFN- γ to induce IDO1 expression, then treated by Mitomycin C prior to mixing with splenocytes. Free EPA, EPA/Lipo-SM/Chol and Epacosome-2 were added to co-culture cells at eq. dose of EPA. To evaluate T cell proliferation, anti-CD3 and IL-2 were added to co-cultures. 3 days later, CD8⁺ T cell proliferation was assessed by FACS analysis.

Extended Data Fig. 2. **a**, Drug administration timeline scheme of healthy C57BL/6 mice received the same treatment as Fig. 4a. **b-f**, On day 15, blood was withdrawn for comprehensive serum chemistry (**b**), leukocytes (**c**), erythrocytes (**d**) and thrombocytes (**e**) analysis, and the liver, spleen, kidneys and muscle were isolated from the mice for hematoxylin & eosin (H&E) staining, and intestinal mucosa were stained by periodic acid-Schiff (PAS) reaction and counterstained with haematoxylin, (**f**). Data are expressed as mean \pm s.d. ($n = 6$). Statistical significance was determined by one-way ANOVA followed by Tukey's multiple comparisons test.

Minor comments

Method

Q1. In the section IDO inhibitory effect in Hela cells, the authors just mentioned “the various concentrations of free EPA or Epacosome-2”. Please mention the concentrations that were used.

Response: Thank you for your suggestion. The concentrations used in the assay were 0, 1, 5, 10, 20, 50, 100, 200 and 10000 nM (equivalent EPA). This information has been included in the **Fig. 2d** legend and the Method section of the revised manuscript.

Q2. In the T cell proliferation assay similar to IDO inhibitory assay, the exact dose of EPA and Epacosome-2 did not mention in the method. Why did not the authors check the proliferation of T cells over a longer period of time? Or in the different time points?

Response: Thank you for spotting this oversight. The doses of EPA and Epacosome-2 used in the T cell proliferation assay were 0, 8, 40, 200, and 1000 nM (equivalent EPA). This information has been included in the **Fig. 2e** legend and the Method section of the revised manuscript. In addition, as suggested by reviewer, we studied the proliferation of T cells over a longer period of time (96 h) and the results were consistent with 72 h (**Supplementary Fig. 8**).

Supplementary Fig. 8. CD8⁺ T cell proliferation by co-culture for 96 h. Data are expressed as mean \pm s.d (n = 3). Statistical significance was determined by one-way ANOVA followed by Tukey's multiple comparisons test.

Q3. In evaluating the therapeutic efficacy of Epacosome-2 in B16-F10 melanoma tumour model, it is better to have a timeline to exactly show the exact time of treatment. This makes it easier to follow for the reader. Also, the same for the healthy mice. The B16-F10 cells injected in DMEM? Or PBS?

Response: We would like to thank the reviewer for the valuable comment. As suggested, we have included the detailed administration scheme in **Figs. 4a,6a,7a,8a, Extended Data Figs. 2a,4a, and Supplementary Figs. 14a,15a,21a**. The B16-F10 cells were injected in serum-free DMED medium. This information has been included in the Method section of the revised manuscript.

Results

Q1. In figure 4, the way that authors used for labeling different groups is confusing. The significance on the tumour growth curve (fig 4c) and survival (fig 4d) results are not clear, especially the tumour growth curve it's hard to follow.

Response: We would like to thank the reviewer for the valuable comment. To avoid confusion, we have further enlarged the font sizes of the labeling and put the full group names as long as the space is allowed in the graphs and included “A, B, C...” as the short designation after each group name. The short designation was used when there is not enough space for the full group names. To allow clearer reading, the original Fig. 4c has been divided into Fig. 4c and Fig. 4d based on with or without α -PD-1 combination. Additionally, the survival curve has been further enlarged. Furthermore, we have performed comprehensive statistical analysis and clearly denoted the significant differences in tumour growth curves (Fig. 4c,d) and survival curves (Fig. 4e). To improve the readability of other graphs in Fig. 4 and avoid crowdedness, the original tumour bearing mice images and representative flow cytometric plots were placed in Supplementary Fig. 12.

Fig. 4. Epacosome-2 potentiated PD-1 blockade in B16-F10 tumour mice via inhibiting IDO1 and eliciting potent CTL responses. **a**, Drug administration timeline scheme of antitumour efficacy in subcutaneous (s.c.) B16-F10 tumour model (n=6, tumours: ~100 mm³), mice injected with Epacosome-2 (i.v.) or free EPA (p.o. or i.v.) at eq. 41 mg EPA/kg

on day 8, 10, 12 and 14 alone or combined with i.p. α -PD-1 (BioXCell, clone RMP1-14, 100 μ g per mouse per 3 day for 3 times) from day 8^{7,8}. **b**, Individual tumour growth curves. **c-d**, Average tumour size growth curves of EPA formulation alone (**c**) or combined with α -PD-1 (**d**). **e**, Kaplan–Meier survival curves. **f-g**, An independent efficacy study in B16-F10 tumour mice (n = 6, tumours: ~100 mm³) as (**a**). **f,g**, The ratio of Trp (nM)/kyn (nM) concentration in plasma (**f**) and tumours (**g**). **h-n**, Quantification analysis of intratumoural CD45⁺/CD11b⁺/F4/80⁺/CD206⁻ M1 and CD45⁺/CD11b⁺/F4/80⁺/CD206⁺ M2 TAMs cells (**h**)^{3,9}, CD45⁺/CD11b⁺/Gr-1⁺ MDSC cells (**i**), CD3⁺/CD4⁺/Foxp3⁺/CD25⁺ Tregs (**j**), CD45⁺/CD11c⁺/CD80⁺/CD86⁺ DCs (**k**), CD45⁺/CD11c⁺/CD103⁺ DCs (**l**), Granzyme B⁺ (**m**) or IFN- γ ⁺/CD8⁺ (**n**) T cells by flow cytometry. Data in **c**, **d**, and **f-n** are expressed as mean \pm s.d. Statistical significance was determined by one-way ANOVA followed by Tukey’s multiple comparisons test; survival curves were compared using the log-rank Mantel–Cox test.

Q2. It is better to present the results of MST as a separate table and analyze the significance between groups. It will also be more informative if the authors present the results of TTE (time to reach endpoint), TGD (tumour growth delay), and ILS (increased live span) for each treatment group and compare the results to show the significant differences.

Response: Thank you for the helpful comment. As requested, we have presented the MST, TTE, TGD and ILS of each treatment group in a separate table (**Supplementary Fig. 13**). The significant differences were determined by one-way ANOVA followed by Tukey’s multiple comparisons test and P values were indicated in the table.

Group (n = 6 mice)	Median survival time (MST, days)	Time to reach endpoint (TTE, days)						Tumour growth delay (TGD, days)						Increased live span (ILS, %)					
		16	18	17	15	22	21	NA	NA	NA	NA	NA	NA	NA	NA	NA	NA	NA	NA
5% Dextrose	17.5	16	18	17	15	22	21	NA	NA	NA	NA	NA	NA	NA	NA	NA	NA	NA	NA
α -PD-1	19	17	20	19	20	19	17	1.2	2.8	2.3	2.8	1.8	1.1	-2.9%	14.3%	8.6%	14.3%	8.6%	-2.9%
EPA (p.o.)	20	19	20	22	20	18	21	2.9	3.4	3.8	2.7	2.7	5.5	8.6%	14.3%	25.7%	14.3%	2.9%	20.0%
EPA (p.o.) + α -PD-1	22.5	19	21	24	23	25	22	2.8	5.2	8.4	1.6	9	6.2	8.6%	20.0%	37.1%	31.4%	42.9%	25.7%
EPA	21.5	25	21	21	22	23	21	6.9	4.8	4.4	5.7	6.9	5	42.9%	20.0%	20.0%	25.7%	31.4%	20.0%
EPA + α -PD-1	24	23	25	26	28	22	22	5.8	9	9	11.2	5.3	6.2	31.4%	42.9%	48.6%	60.0%	25.7%	25.7%
Epacosome-2	27	26	27	29	27	26	28	10.9	12	13.5	11.3	11.4	13	48.6%	54.3%	65.7%	54.3%	48.6%	60.0%
Epacosome-2 + α -PD-1	31	30	29	31	32	31	34	15	13.7	15	16.7	16	18.4	71.4%	65.7%	77.1%	82.9%	77.1%	94.3%
Epacosome-2 + α -CD8a	18.5	22	18	21	17	16	19	4.9	0.3	0.8	0.7	0.9	0	25.7%	2.9%	20.0%	-2.9%	-8.6%	8.6%
Epacosome-2 + α -PD-L1 + α -CD8a	19	23	17	19	19	20	18	2.9	2.3	3.6	0.4	0.3	3.3	31.4%	-2.9%	8.6%	14.3%	2.9%	
Statistical significance comparison		P value by one-way ANOVA followed by Tukey’s multiple comparisons test																	
EPA (p.o.) vs Epacosome-2		<0.0001						<0.0001						<0.0001					
EPA vs Epacosome-2		0.0021						<0.0001						0.0004					
EPA (p.o.) + α -PD-1 vs Epacosome-2 + α -PD-1		<0.0001						<0.0001						<0.0001					
EPA + α -PD-1 vs Epacosome-2 + α -PD-1		<0.0001						<0.0001						<0.0001					
Epacosome-2 vs Epacosome-2 + α -PD-1		0.0286						0.0063						0.0089					

Supplementary Fig. 13. A table shows the median survival time (MST), time to reach endpoint (TTE), tumour growth delay (TGD) and increased live span (ILS) from **Fig. 4**. Statistical significance was determined by one-way ANOVA followed by Tukey’s multiple comparisons test.

Q3. Since melanoma is an aggressive type of cancer and induction of an effective immune response is hard to achieve, it will be also valuable to check the level of exhausted CD8 T cells (i.e., PD-1, Tim-3, and Lag-3).

Response: We would like to thank the reviewer for the valuable comment. As suggested, we have analyzed the intratumoural exhausted CD8⁺ T after the tumour bearing mice received the same treatments as in **Fig. 6 (Extended Data Fig. 1)**. We showed that the levels of CD8⁺/PD-1⁺ exhausted T cells were significantly increased in groups containing DTIC/Lipo, especially in the

DTIC/Epacosome-2. However, no significant differences of CD8⁺/LAG-3⁺ and CD8⁺/Tim-3⁺ T cells were observed after the treatments. These data justified the combination with α-PD-1 therapy to further improve the antitumour efficacy.

Extended Data Fig. 1 (partial). Representative flow cytometric plots of intratumoural PD-1⁺, Lag-3⁺ and Tim-3⁺ CD8⁺ T cells their respective quantification (right panel, 3 tumours were randomly chosen from an independent assay on day 18 after receiving the same treatments as Fig. 6a). Data are expressed as mean ± s.d. Statistical significance was determined by one-way ANOVA followed by Tukey's multiple comparisons.

Q4. In presenting the results of figure 6 in the body of the article, the author presented graphs a-e and then jumped to g-l and again f, m. Please keep them in order.

Response: Thank you for spotting this oversight. We have revised the manuscript accordingly to keep the Fig. 6 graphs discussed in order.

Q5. In Figure 7, the DTIC/Epacosome-2 showed the smallest tumour size among all treatment groups, but the significance in fig 7b is not clear. Does it mean that all the groups were compared to DTIC/Epacosome-2 group?

Response: Thank you for the comments and sorry for the confusion. To make it clearer, we have also compared all groups to DTIC/Epacosome-2 and relevant P values were presented.

Fig. 7c (original Fig. 7b). Tumour growth curves measured by a digital caliper. Data **c** are expressed as mean \pm s.d. Statistical significance was determined by one-way ANOVA followed by Tukey's multiple comparisons test.

Q6. Again, the significance in figure 8c, d is not clear. The table of TTE, TGD, MST, and ILS results also works for the combination therapy with PD-1 mAb study.

Response: Thank you for the critical comment. As requested, the exact P values have been indicated in **Fig. 8c,d** (now **Fig. 8d,e**) in the revised manuscript. We have also presented the MST, TTE, TGD and ILS in the **Supplementary Fig. 22** and the significant differences (P values) for each comparison were also included in the table.

Fig. 8d,e (original Fig. 8c,d). **d**, Tumour growth curves measured by a digital caliper. **e**, Kaplan–Meier survival curves. Data in **d** are expressed as mean \pm s.d. Statistical significance in **d** was determined by one-way ANOVA followed by Tukey's multiple comparisons test; survival curves were compared using the log-rank Mantel–Cox test.

Group (n = 5 mice)	Median survival time (MST, days)	Time to reach endpoint (TTE, days)					Tumour growth delay (TGD, days)					Increased live span (ILS, %)				
		21	24	23	21	22	NA	NA	NA	NA	NA	NA	NA	NA	NA	NA
5% Dextrose	22	21	24	23	21	22	NA	NA	NA	NA	NA	NA	NA	NA	NA	NA
α -PD-1	23	24	22	23	24	23	3.2	1.8	0.7	1	2.4	9.1%	0.0%	4.5%	9.1%	4.5%
DTIC/Lipo-SM/Chol + Epacosome-2	27	24	26	27	28	29	6.2	3	4.2	2.3	2.7	27.3%	18.2%	31.8%	9.1%	22.7%
DTIC/Epacosome-2	30	28	29	30	31	32	8	6.9	6.7	6.5	5.6	45.5%	40.9%	36.4%	31.8%	27.3%
DTIC/Lipo-SM/Chol + Epacosome-2 + α -PD-1	31	27	28	32	31	33	8	7.5	9.3	8.9	6.5	40.9%	27.3%	50.0%	45.5%	22.7%
DTIC/Epacosome-2 + α -PD-1	35	35	36	37	32	35	14	12.8	11.6	15.8	12.2	63.6%	59.1%	45.5%	68.2%	59.1%
Statistical significance comparison		P value by one-way ANOVA followed by Tukey's multiple comparisons test														
DTIC/Lipo-SM/Chol + Epacosome-2 vs DTIC/Epacosome-2		0.0817					0.0194					0.0608				
DTIC/Lipo-SM/Chol + Epacosome-2 + α -PD-1 vs DTIC/Epacosome-2 + α -PD-1		0.0031					<0.0001					0.0018				
DTIC/Epacosome-2 vs DTIC/Epacosome-2 + α -PD-1		0.0020					<0.0001					0.0012				

Supplementary Fig. 22. A table shows the median survival time (MST), time to reach endpoint (TTE), tumour growth delay (TGD) and increased live span (ILS) from Fig. 8. Statistical significance was determined by one-way ANOVA followed by Tukey's multiple comparisons test.

Q7. Please clarify which day was considered as the end of the study in combination therapy study. In the individual growth curve in each group (fig 8b) day 22 was shown, in tumour size (fig 8c) day 32 and in survival curve (fig 8d) day 37? Since all mice in both dextrose and ant PD-1 groups were died by day 22, the author can use border line on day 22 in fig 8b for all treatment groups to show the number and the tumour growth of mice in treatment groups till end of study that seems to be day 37.

Response: We would like to thank the reviewer for the valuable comment. The endpoint for each mouse was when tumour size was found to be $\geq 2000 \text{ mm}^3$. To avoid confusion, the individual tumour growth curves have been updated to the last day for each mouse to match the survival curves. For the average tumour growth curves, as suggested by reviewer, the end of the study was chosen to be at day 22 as this was when some tumours in vehicle control group reached $\geq 2000 \text{ mm}^3$.

Fig. 8c-e. Individual (c) and average (d) tumour growth curves measured by a digital caliper. e, Kaplan–Meier survival curves. Data in d are expressed as mean \pm s.d. Statistical significance in d was determined by one-way ANOVA followed by Tukey's multiple comparisons test; survival curves were compared using the log-rank Mantel–Cox test.

Q8. It would be noteworthy if the authors could do IHC or mIF for visualizing the infiltrated T cells and reduced Tregs, NK cells, ... other immune cells in the tumour site of different treatment groups.

Response: Thank you for the constructive comments. As requested, the IHC of CD8, Granzyme B, IFN- γ , perforin, Foxp3, NKG2D, CD69 and NK1.1 in tumours (**Fig. 6b**) have been performed and the results were consistent with the flow cytometry studies (**Supplementary Fig. 19e**).

Supplementary Fig. 19e. Representative IHC staining for CD8, Granzyme B, IFN- γ , perforin, Foxp3, NKG2D, CD69 and NK1.1 in tumours from **Fig. 6** on day 18, scale bar = 100 μ m.

Discussion

It is more Like the results part and used figures. The authors should discuss more and compare with previous studies.

Response: Thank you for the comments. As suggested, we have updated the discussion part via adding the comparison of our nanosystem with previous studies in the revised manuscript as following:

Moreover, comparing with physically encapsulating EPA in liposomes, including the reported EPA/Lipo-DOPE/Chol and EPA/Lipo-HSPC/Chol^{1,2}, Epacosome possessed significantly higher EPA DLC (8.19-16.38% vs 0.23-1.27%, **Figs. 1a** and **Supplementary Fig. 5c**). Also, Epacosome-2 markedly improved the pharmacokinetics by extending the circulation half-life (4.72 vs 0.72 h) and delivering more EPA into the tumour (4.17 vs 1.05%) with better tumour penetration efficiency as compared to physically loading EPA into the liposome consisting of the SM/Chol/DSPE-PEG_{2K} (EPA/Lipo-SM/Chol, **Figs. 3**). The improved pharmacokinetics and tumour distribution could be attributed to the enhanced formulation stability of Epacosome-2 (**Supplementary Fig. 5a,b,e,f**).

Furthermore, Epacosome-2 performed significantly better than EPA/Lipo-SM/Chol, EPA/Lipo-DOPE/Chol and EPA/Lipo-HSPC/Chol on anti-melanoma efficacy and immune responses (**Supplementary Figs. 14,15**).

Reviewer #3:

Main points/questions:

Q1. Fig 2: B16 cells don't express high IDO1. Please use alternative tumours such as 4T1 or B16 that overexpress IDO1 to validate claim of epacosome mediated IDO1 inhibition.

Response: We would like to thank the reviewer for the valuable comment. We agree that B16 cells don't overexpress IDO1, and that's why in **Fig 2d** we used Hela cells for the IDO1 enzymatic assay based on the literature.³ In addition, as suggested by reviewer, we have also performed this assay in 4T1 cells, and the results were consistent with **Fig. 2d**, in which Epacosome-2 showed the highest level of IDO1 inhibition (**Supplementary Fig. 9d**).

Supplementary Fig. 9d, IDO1 enzymatic inhibitory activity via measuring the Kyn in supernatants in 4T1 cells treated with IFN- γ along with free EPA, Lipo-SM/Chol, EPA/Lipo-SM/Chol and Epacosome-2 at equivalent (eq.) EPA concentration for 48 h. Data are expressed as mean \pm s.d (n = 3). Statistical significance was determined by one-way ANOVA followed by Tukey's multiple comparisons test.

Q2. Fig 2: Authors show different epacosome iterations but do not have a negative control (with empty nanovesicle or non-active compound) in addition to epacosome. This is needed to differentiate effect of nanovesicle affect over the delivery of epacadostat.

Response: We would like to thank the reviewer for the valuable comment. We agree with reviewer on setting empty liposome as a negative control to differentiate effect of nanovesicle over the delivery of epacadostat. To that end, we have prepared the empty liposome, Lipo-SM/Chol composed of the SM/Chol/DSPE-PEG_{2K} with the same lipid ratio as Epacosome-2 (SM in EPA/Lipo-SM/Chol = "SM + SM-EPA" in Epacosome-2, **Supplementary Fig. 5c**). We demonstrated that the empty liposome, Lipo-SM/Chol had no effect on IDO1 inhibition and did not render any benefits for CD8⁺ T cells proliferation (**Fig. 2d,e**). Furthermore, compared to vehicle control, Lipo-SM/Chol exerted negligible inhibition on the tumour growth in B16-F10 tumour model and had no significant effects on the Trp/Kyn ratio in both plasma and tumours as well as the antitumour immune responses (**Supplementary Fig. 14**). In sum, these findings demonstrated that the observed antitumour effects of Epacosome-2 were not related to the empty nanovesicle.

C

Liposomes	Lipid ratio	EPA loading efficiency (%)	EPA loading capacity (weight %)	DLS Size by intensity (d.nm)	Zeta Potential (mV)	Polydispersity Index (PDI)
Lipo-SM/Chol (empty liposome)	SM:Chol:DSPE-PEG _{2K} = 82.82/12.78/4.40 (molar ratio)	NA	NA	120.2 ± 3.06	-10.3 ± 2.65	0.110 ± 0.024
EPA/Lipo-SM/Chol	SM:Chol:DSPE-PEG _{2K} = 82.82/12.78/4.40 (molar ratio)	72.3	1.27	154.5 ± 7.53	-18.5 ± 4.83	0.114 ± 0.016
EPA/Lipo-DOPE/Chol	DOPE:DOTAP:Chol:DSPE-PEG _{2K} = 9/9/2/1 (weight ratio)	62.1	1.08	126.6 ± 4.98	-16.6 ± 6.82	0.228 ± 0.018
EPA/Lipo-HSPC/Chol	HSPC:Chol:DSPE-PEG _{2K} = 22.3/7.3/7.3 (weight ratio)	53.6	0.23	117.7 ± 5.03	-12.6 ± 3.67	0.115 ± 0.010

Supplementary Fig. 5c, A table illustrating the physicochemical properties of Lipo-SM/Chol (empty liposome), EPA/Lipo-SM/Chol, EPA/Lipo-DOPE/Chol and EPA/Lipo-HSPC/Chol with detailed ratios^{1,2}. d.nm, diameter values in nanometres. DLS, dynamic light scattering.

Fig. 2d,e, **d**, IDO1 enzymatic inhibitory activity via measuring the Kyn in supernatants in HeLa cells treated with IFN- γ along with free EPA, EPA/Lipo-SM/Chol and Epacosome-2 at equivalent (eq.) EPA concentration for 48 h. **e**, T cell proliferation by co-culture^{3,4}. B16-F10 cells were stimulated by IFN- γ to induce IDO1 expression, then treated by Mitomycin C prior to mixing with splenocytes. Free EPA, EPA/Lipo-SM/Chol and Epacosome-2 were added to co-culture cells at eq. dose of EPA. To evaluate T cell proliferation, anti-CD3 and IL-2 were added to co-cultures. 3 days later, CD8⁺ T cell proliferation was assessed by FACS analysis. Data are expressed as mean \pm s.d. (n = 3). Statistical significance was determined by one-way ANOVA followed by Tukey's multiple comparisons test.

Supplementary Fig. 14. a, Drug administration timeline scheme of antitumour efficacy in subcutaneous (s.c.) B16-F10 tumour model (n=6, tumours: ~100 mm³), mice i.v. injected with Lipo-SM/Chol (empty liposome), EPA/Lipo-SM/Chol, Epacosome-2 at eq. 41 mg EPA/kg on day 8, 10, 12 and 14 alone or combined with i.p. α -PD-1 (BioXCell, clone RMP1-14, 100 μ g per mouse per 3 day for 3 times) from day 8. **b**, Individual tumour growth curves. **c-d**, Average tumour size growth curves (**c**) and body weight (**d**). **e,f**, The ratio of Trp (nM)/kyn (nM) concentration in plasma (**e**) and tumours (**f**). **g**, Mice bearing s.c. B16-F10 tumour images taken on day 15. **h**) Representative IHC staining for CD8, Granzyme B, IFN- γ , and Foxp3 on day 15, scale bar = 100 μ m. Data in **c-f** are expressed as mean \pm s.d. Statistical significance was determined by one-way ANOVA followed by Tukey's multiple comparisons test.

Q3. Fig 3: Liver expresses high levels of IDO1 and TDO2. There is an increase in liver and spleen epacosome uptake more so than other organs. Tissue damage including looking at necropsy, is needed to assess potential toxicity.

Response: Thank you for the constructive comment. To assess potential toxicity in liver, spleen and other organs, we have investigated histopathology in various tissues in healthy mice that received the same treatments as in **Fig. 4a**. Our results showed that Epacosome-2 did not induce overt toxicity in liver, spleen, kidney, muscle, and intestinal mucosa, corroborating the excellent *in vivo* safety profiles of Epacosome-2.

Extended Data Fig. 2f. The liver, spleen, kidneys, and muscle were isolated from the mice for hematoxylin & eosin (H&E) staining, and intestines were stained by periodic acid-Schiff (PAS) reaction and counterstained with haematoxyli. Representative H&E staining of liver, spleen, kidneys and muscle, and PAS staining of intestine mucosa (f).

Q4. Extended Data 3 and 5: Authors need to reanalyze flow cytometry data to include flow plots that gate on live cells and subsequent distinct population gating in addition to FMO. The concern is nonviable cells and debris is being gated on that skew analysis. It would probably important to seek the expertise of someone who has deeper expertise in flow cytometry to help with this aspect.

Response: We would like to thank the reviewer for the valuable comment. As suggested by reviewer, with the assistance of the Flow Cytometry Immune Monitoring Shared Resource

(FCIMSR) core at UArizona Cancer Center, we have used Zombie violet to gate on live cells, and performed FMO after subsequent distinct population gating to reanalyze all the flow cytometry data. The results have been updated in Fig. 6h-n, Extended Data Figs. 1,3,5,6 and Supplementary Fig. 12b.

Extended Data Fig. 3. Gating strategy used to define intratumoural (CD45⁺/CD11b⁺/F4/80⁺/CD206[±]) TAMs cells, (CD45⁺/CD11b⁺/Gr-1⁺) MDSC cells, (CD4⁺/CD3⁺/CD25⁺/Foxp3⁺) Tregs, (CD45⁺/CD11c⁺/CD80⁺/CD86⁺) and (CD45⁺/CD11c⁺/CD103⁺) DCs, (CD45⁺/Rae-1⁺), (CD45⁺/Mult-1⁺) and (CD45⁺/MHC-I⁺) tumour cells.

Extended Data Fig. 5. Gating strategy used to define intratumoural (CD3⁺/NK1.1⁺/NKG2D⁺), (CD3⁺/NK1.1⁺/CD69⁺), (CD3⁺/NK1.1⁺/IFN- γ ⁺), (CD3⁺/NK1.1⁺/perforin⁺) and (CD3⁺/NK1.1⁺/granzyme B⁺) NK cells.

Extended Data Fig. 6. Gating strategy used to define intratumoural (CD3⁺/CD8⁺/NKG2D⁺), (CD3⁺/CD8⁺/CD69⁺), (CD3⁺/CD8⁺/IFN- γ ⁺), (CD3⁺/CD8⁺/perforin⁺), (CD3⁺/CD8⁺/granzyme B⁺), (CD3⁺/CD8⁺/PD-1⁺), (CD3⁺/CD8⁺/Lag-3⁺) and (CD3⁺/CD8⁺/Tim-3⁺) T cells.

Q5. Figure 6B: Subplot in tumour measurements does not have X and Y labels and not clear as to what the plot is referring to.

Response: We would like to thank the reviewer for the valuable comment. We have presented the X and Y labels as “Days after B16-F10-Luc2 cells inoculation” and “Tumour size (mm³)” in the **Fig. 6b** subplot, respectively. This subplot refers to the same F, G, H groups presented in the parental **Fig. 6b**. To make the whole figure clearer, the full group names were added to the updated **Fig. 6b**.

Fig. 6b, Average tumour growth curves measured by a digital caliper. Data are expressed as mean \pm s.d. Statistical significance was determined by one-way ANOVA followed by Tukey’s multiple comparisons.

Q6. Fig 7: Confirmation of depletion of immune cell subsets needs to be addressed.

Response: We would like to thank the reviewer for the valuable comment. We have performed IHC analysis for the intratumoural CD8, IFN- γ , NKG2D and NK1.1 after injecting the α -CD8a, α -IFN- γ , α -NKG2D, or α -NK1.1 antibodies to mice (**Supplementary Fig. 20b**). Our results confirmed the depletion of the corresponding immune cell subsets.

Supplementary Fig. 20b. Representative IHC staining of intratumoural CD8, IFN- γ , NKG2D and NK1.1 from **Fig. 7** on day 17, scale bar = 100 μ m.

Minor points

Q1. Fig 2: Use macrophage for uptake in addition to tumour.

Response: Thank you for the constructive comment. As requested, we have performed the cellular uptake in macrophage RAW264.7 cells (**Supplementary Fig. 9b**). Consistently, Epacosome-2 had higher cellular uptake than free EPA and EPA/Lipo-SM/Chol in RAW264.7 cells.

Supplementary Fig. 9b. Cellular uptake levels of Epacosome-2 in RAW 264.7 cells after 24 and 48 h incubation.

Q2. Fig 4: panel B and G is unnecessary remove to supplement keep the summary bar plots.

Response: Thank you for the comment. As suggested, the panel b and g have been moved to **Supplementary Fig. 12** and the summary bar plots have been kept in **Fig. 4**.

Q3. Reference to Figures is out of order of appearance.

Response: Thank you for spotting this oversight. We have revised the manuscript accordingly to keep the reference to figures in order of appearance.

Q4. Fig 6: Missing survival data

Response: Thank you for the comment. Kaplan-Meier survival curves were collected from an independent efficacy study as **Fig. 6 (Supplementary Fig. 19d)**. DTIC/Epacosome-2 prolonged mouse survival rate compared to DTIC/Lipo-SM/Chol + Epacosome-2, particularly when combined with α -PD-1. To avoid the crowdedness of **Fig. 6**, the full panel of mouse bioluminescence imaging has been moved to the **Supplementary Fig. 19a**, leaving a mouse in each group as a representative.

Supplementary Fig. 19d. Kaplan-Meier survival curves from an independent efficacy study in B16-F10 tumour mice (n=6, tumours: ~300 mm³) after receiving the same treatment as **Fig. 6**. Survival curves were compared using the log-rank Mantel-Cox test.

5. Justification for anti-PD1 at 100ug dose typically 200ug – 250ug and reference clone used.

Response: Thank you for the comment. α -PD-1 (BioXCell, clone RMP1-14) at 100 μ g/mouse was used according to the published literature^{7,8}. We have updated the references and clone in the revised manuscript.

6. More rigor is needed in figure preparation and paper write up. Multiple grammar and misspelling make reading manuscript difficult, careful editing needed. Misspelling of epacosome, MDSC, etc. This is unfortunate as the premise of this work is great.

Response: We would like to thank the reviewer for the valuable comment. We have scrutinized the manuscript for grammar and misspelling and have corrected related mistakes accordingly in the revised manuscript.

7. Multiple figures of tumour bearing image of mice (Fig. 4B, 6D, 8A) need to be shown in qualitative measurements with representative images placed in supplement.

Response: We would like to thank the reviewer for the valuable comment. For tumour bearing image of mice, original **Fig. 4b** has been moved to **Supplementary Fig. 12a**, original **Fig. 6a** has been moved to **Supplementary Fig. 19a** with one representative mouse image left in **Fig. 6c**. Original **Fig. 6d** was heatmap of tumour metastatic rate, and original **Fig. 8a** was mouse bioluminescence imaging. The related quantitative measurements of tumours are presented in **Figs. 4c,d,6b,8d**.

We hope that all the raised concerns have been addressed in a satisfactory manner and that this revised manuscript is now acceptable for publication in *Nature Communications*. Please let us know should more information is needed. Thank you very much for your consideration. We wish you and your loved ones are safe and healthy during this troubling time.

References

- 1 Chen, Y., Du, Q., Zou, Y., Guo, Q., Huang, J., Tao, L., Shen, X. & Peng, J. Co-delivery of doxorubicin and epacadostat via heparin coated pH-sensitive liposomes to suppress the lung metastasis of melanoma. *International journal of pharmaceutics* **584**, 119446 (2020).
- 2 Tahaghoghi-Hajghorbani, S., Khoshkhabar, R., Rafiei, A., Ajami, A., Nikpoor, A. R., Jaafari, M. R. & Badiee, A. Development of a novel formulation method to prepare liposomal Epacadostat. *European journal of pharmaceutical sciences : official journal of the European Federation for Pharmaceutical Sciences* **165**, 105954 (2021).
- 3 Chen, Y., Xia, R., Huang, Y., Zhao, W., Li, J., Zhang, X., Wang, P., Venkataramanan, R., Fan, J., Xie, W., Ma, X., Lu, B. & Li, S. An immunostimulatory dual-functional nanocarrier that improves cancer immunochemotherapy. *Nature communications* **7**, 13443 (2016).
- 4 Parrish, H. L., Deshpande, N. R., Vasic, J. & Kuhns, M. S. Functional evidence for TCR-intrinsic specificity for MHCII. *Proceedings of the National Academy of Sciences of the United States of America* **113**, 3000-3005 (2016).
- 5 Long, G. V., Dummer, R., Hamid, O., Gajewski, T. F., Caglevic, C., Dalle, S., Arance, A., Carlino, M. S., Grob, J. J., Kim, T. M., Demidov, L., Robert, C., Larkin, J., Anderson, J. R., Maleski, J., Jones, M., Diede, S. J. & Mitchell, T. C. Epacadostat plus pembrolizumab versus placebo plus pembrolizumab in patients with unresectable or metastatic melanoma (ECHO-301/KEYNOTE-252): a phase 3, randomised, double-blind study. *Lancet Oncol* **20**, 1083-1097 (2019).
- 6 Sanborn, G. E., Niederkorn, J. Y. & Gamel, J. W. Efficacy of dacarbazine (DTIC) in preventing metastases arising from intraocular melanomas in mice. *Graefes Arch Clin Exp Ophthalmol* **230**, 192-196 (1992).
- 7 Edwards, S. C., Hedley, A., Hoevenaar, W. H. M., Wiesheu, R., Glauner, T., Kilbey, A., Shaw, R., Boufea, K., Batada, N., Hatano, S., Yoshikai, Y., Blyth, K., Miller, C., Kirschner, K. & Coffelt, S. B. PD-1 and TIM-3 differentially regulate subsets of mouse IL-17A-producing gammadelta T cells. *The Journal of experimental medicine* **220** (2023).
- 8 Wang, Z., Little, N., Chen, J., Lambesis, K. T., Le, K. T., Han, W., Scott, A. J. & Lu, J. Immunogenic camptothosome nanovesicles comprising sphingomyelin-derived camptothecin bilayers for safe and synergistic cancer immunochemotherapy. *Nat. Nanotechnol.* **16**, 1130-1140 (2021).
- 9 Chen, Y., Huang, Y., Li, Q., Luo, Z., Zhang, Z., Huang, H., Sun, J., Zhang, L., Sun, R., Bain, D. J., Conway, J. F., Lu, B. & Li, S. Targeting Xkr8 via nanoparticle-mediated in situ co-delivery of siRNA and chemotherapy drugs for cancer immunochemotherapy. *Nature nanotechnology* **18**, 193-204 (2023).

REVIEWERS' COMMENTS

Reviewer #1 (Remarks to the Author):

The authors have addressed my comments.

Reviewer #3 (Remarks to the Author):

We thank the authors and appreciate their efforts for attempting to answer all our comments.

There are still some lingering limitations and they are as follows:

- Flow plots are hard to read and pixelated. Improving them is important for the readership.
- Figure' labels are missing and inconsistent.
- A few of the figures are pixelated including the histology figures which makes it difficult to evaluate.
- The premise of the paper is sound. However, the efficacy of the SM-EPA nanoparticle remains a bit unclear.

Reviewer #1: (Remarks to the Author):

The authors have addressed my comments.

Response: Thank you!

Reviewer #3: (Remarks to the Author):

We thank the authors and appreciate their efforts for attempting to answer all our comments.

There are still some lingering limitations and they are as follows:

Q1. Flow plots are hard to read and pixilated. Improving them is important for the readership.

Response: Thank you for the critical comment. We agree with the reviewer on the necessity of improving the resolution of the flow plots for the readership. As required by the editorial office, we have uploaded the high-resolution flow plots as a separate figure file for Fig. **6h-n** to the Manuscript Tracking System. In addition, the flow plots in **Supplementary Figure 12, 19-22 (Extended Data Fig. 1,3,5,6** in the old version) have also been updated to high-resolution format in supplementary information.

Q2. Figure' labels are missing and inconsistent.

Response: Thank you for spotting this oversight. We have added the labels in the **Supplementary Figures 20-22** and updated the legends accordingly. Additionally, to be consistent, above the figures of "Hematocrit" in **Supplementary Figures 13,23, (Extended Data Figs. 2,4** in the old version) "Erythrocytes" labeling was added", and the "EPA" has been replaced with "Free EPA" in the **Supplementary Figure 7**.

Q3. A few of the figures are pixilated including the histology figures which makes it difficult to evaluate.

Response: We would like to thank the reviewer for the valuable comment. As required by the editorial office, we have uploaded the high-resolution figure files for **Figs. 1-8** to the Manuscript Tracking System. Furthermore, the figures in Supplementary Information have also been updated to high-resolution format.

Q4. The premise of the paper is sound. However, the efficacy of the SM-EPA nanoparticle remains a bit unclear.

Response: Thank you for your comment. Epacosome-2 outperformed free EPA (p.o. or i.v.) or α -PD-1 in reducing the B16-F10 tumour growth (**Figs. 4b-c and Supplementary Figure 12a**). Furthermore, Epacosome-2 beat free EPA (p.o. or i.v.) in enhancing the efficacy of α -PD-1 as demonstrated by the further delayed tumour development and prolonged mouse survival (**Fig. 4b-d and Supplementary Figure 12a**). Systematic analysis of the median survival time (MST), time to reach endpoint (TTE), tumour growth delay (TGD), and increased live span (ILS) corroborated the therapeutic benefits of Epacosome-2, particularly when combined with α -PD-1 (**Supplementary Table 1**). The fortified anti-melanoma activity of Epacosome-2 could be attributed to the improved pharmacokinetics, and the bolstered tumour accumulation and penetration (**Fig. 3**).

We hope that all the raised concerns have been addressed in a satisfactory manner and that this revised manuscript is now acceptable for publication in Nature Communications. Please let us know should more information is needed. Thank you very much for your consideration.

We wish you and your loved ones are safe and healthy during this troubling time.